# Motion model ultrasound localization microscopy for preclinical and clinical multiparametric tumor characterization

Tatjana Opacic[1], Stefanie Dencks [2], Benjamin Theek[1], Marion Piepenbrock[2], Dimitri Ackermann[2], Anne Rix[1], Twan Lammers[1], Elmar Stickeler[3], Stefan Delorme[4], Georg Schmitz [2] & Fabian Kiessling[1]

Super-resolution imaging methods promote tissue characterization beyond the spatial resolution limits of the devices and bridge the gap between histopathological analysis and non-invasive imaging. Here, we introduce motion model ultrasound localization microscopy (mULM) as an easily applicable and robust new tool to morphologically and functionally characterize fine vascular networks in tumors at super-resolution. In tumor-bearing mice and for the first time in patients, we demonstrate that within less than 1 min scan time mULM can be realized using conventional preclinical and clinical ultrasound devices. In this context, next to highly detailed images of tumor microvascularization and the reliable quantification of relative blood volume and perfusion, mULM provides multiple new functional and morphological parameters that discriminate tumors with different vascular phenotypes. Furthermore, our initial patient data indicate that mULM can be applied in a clinical ultrasound setting opening avenues for the multiparametric characterization of tumors and the assessment of therapy response.

[1] Institute for Experimental Molecular Imaging, University Clinic Aachen, RWTH Aachen University, CMBS, Forckenbeckstr. 55, 52074 Aachen, Germany. [2] Chair for Medical Engineering, Department of Electrical Engineering and Information Technology, Ruhr University Bochum, Universitätsstr. 150, 44780 Bochum, Germany. [3] Department of Obstetrics and Gynecology, University Clinic Aachen, RWTH Aachen University, Pauwelsstr. 30, 52074 Aachen, Germany. [4] Department of Radiology, German Cancer Research Center, Im Neuenheimer Feld 280, 69120 Heidelberg, Germany. These authors contributed equally: Tatjana Opacic, Stefanie Dencks. Correspondence and requests for materials should be addressed to G.S. (email: georg.schmitz@rub.de) or to F.K. (email: fkiessling@ukaachen.de)

Ultrasound (US) is among the most frequently used diagnostic modalities in clinical routine, and its spatial and temporal resolution as well as tissue contrast have been steadily improved. The application of gas-filled microbubbles (MB) as US contrast agents further enhances the diagnostic accuracy of US by adding morphological and functional information about the tissue vascularization[1]. This is particularly relevant in oncology, since the vascular structure of tumors contains essential information for their differential diagnosis[2–4], prognostication[5], and for the prediction and monitoring of therapy responses[6–8]. In particular, some vascular features have already been shown to be capable of identifying patients not responding to antiangiogenic therapy[9], who, then, can be reoriented toward alternative approaches[10,11].

Different qualitative and quantitative techniques have been developed to extract the information about tumor vasculature contained in contrast-enhanced US (CEUS) scans. However, in state-of-the-art CEUS imaging, e.g., using maximum intensity over time (MIOT)[12] or replenishment kinetics analysis[13], voxels are usually much larger than the majority of tumor blood vessels, whose diameters are in the range of 5–80 μm[14]. This limitation in the spatial resolution makes it difficult to gain a comprehensive overview of the vascular architecture and its heterogeneity. In addition, since the probability is high that every voxel contains at least one blood vessel, the tumor vascularization tends to be overestimated whenever the relative blood volume (rBV) is determined based on the area that exhibits MB signals[15]. Voxel-wise analyses are further complicated by high background noise, which can make the assessment of functional vascular parameters difficult and unreliable at the single voxel level[16].

To overcome these issues, several postprocessing algorithms for CEUS image analysis have recently been proposed to reveal and quantify vascular features at super-resolution, which means at a resolution beyond the resolution limits of the device[17,18]. Here, individual MB are localized, and a line with the thickness of a MB is drawn connecting the most closely localized MB in two subsequent frames. This line represents the track of a MB and thus, the course of a (micro) vessel. The approach was successfully applied to characterize MB flow tracks in brain[17] and ear vessels[18]. However, in case of ambiguous assignment possibilities, this approach could lead to underestimation of flow velocities and might be particularly difficult to apply to more complex tumor vascular networks. Therefore, Errico and colleagues[17] used an experimental imaging system with a very high frame rate (500 frames per second). By detecting the moving MB more frequently, ambiguous assignments are avoided and the overall detection probability for a MB rises. Additionally, in-plane motion estimation and correction are improved. However, while the detection probability of MB is increased, the total number of MB available in the image slice within a given acquisition time is determined by the contrast agent blood concentration and the flow-rate of MB in the vessels. Thus, the higher frame rates improve the overall image quality and the correct localization of the vessels' course but not the total number of detected vessels. Furthermore, comparable frame rates are not realized in the majority of clinical US systems so far, which makes clinical translation of this method difficult at the moment.

Therefore, we present here an alternative super-resolution CEUS approach called motion model ultrasound localization microscopy (mULM), which is an advanced tracking technique that is adapted to clinical settings. With mULM, within less than a minute and using a conventional US device operating at standard frame rates, super-resolution images and novel parameters can be extracted, which enable an accurate discrimination of tumors with different vascular phenotypes. Furthermore, the preliminary clinical data that are presented in this manuscript show that rapid translation of mULM is realistic and that this technology may improve the diagnostic potential of CEUS in future clinical practice.

## Results

**Structural imaging of tumor vasculature with mULM.** The mULM method reliably captured the movement of MB in tumors and could be successfully applied to all contrast-enhanced scans using a commercial US device operating at frame rates of ~50 frames per second. It only required measurement times of 40 s (for more information about the algorithm, see Fig. 1a as well as the detailed description in the Methods section). A spatial resolution of ~10 μm was achieved as we determined by measuring the extent of the smallest vessels flown through by more than five MB in one pixel (see Supplementary Figure 1).

The vascular architecture of tumors was visualized in fine detail (Fig. 1b). Functional parameters were calculated for single vessels and combined with textural features, which so far could not be obtained with standard CEUS imaging (Fig. 1c). This enabled us to depict differences in the vascular texture of different tumor models at super-resolution. For instance, in line with histological stainings, the highly angiogenic A431 tumors showed a fine network of very small vessels, homogeneously distributed throughout the entire tumor tissue (Figs. 1b, 2). In contrast, A549 tumors, which are known to be less vascularized and characterized by a more mature vascular system, displayed a higher vascular hierarchy in super-resolution mULM images, with larger vessels at the periphery, that branched into smaller vessels toward the tumor center. MLS tumors with their heterogeneous vascular pattern were most difficult to classify. In the super-resolution mULM images, as in histology, they were characterized by highly and poorly vascularized regions, and by more or less dense and branched vascular areas (Fig. 2).

As an additional feature of the mULM approach, velocities of MB and their directions of movement can be calculated for individual vessels and displayed in parametric maps (Figs. 1b, 2b, c). In these parametric direction maps information about arterial and venous supply as well as branching and connections between vessels is provided. While the velocity profiles of the tumors were similar, the analysis of flow directions showed differences. The parametric direction maps of A431 tumors indicated that in the tumor periphery there were more and better organized arterial than veneous vessels, while in the tumor center the tumor blood flow tended to be directed toward the periphery but with only rare connections to large veins (Fig. 1b, Supplementary Figure 2, and Supplementary Table 1). This lack of venous tumor drainage is known to be a typical characteristic of highly angiogenic tumors[19,20]. In contrary, the parametric maps of A549 and MLS tumors, which have a more mature vascularization (Supplementary Figure 3), showed a balanced mixture of feeding and draining vessels (Fig. 2c), which is also reflected by their higher local flow direction entropy values (Fig. 3b).

**Characterization of vascular tumor phenotypes using mULM.** While some mULM parameters can also be obtained by state-of-the-art CEUS postprocessing techniques, others represent new parameter classes that so far have been difficult to assess (Fig. 1c and Supplementary Table 2). Parameters determined by mULM include the relative blood volume (rBV), the mean, variance, maximum and median values of MB velocities, distances to the closest vessel, and distances to vessel with low and high velocities, as well as the local flow direction entropy as a measure for the organization of the vessel networks.

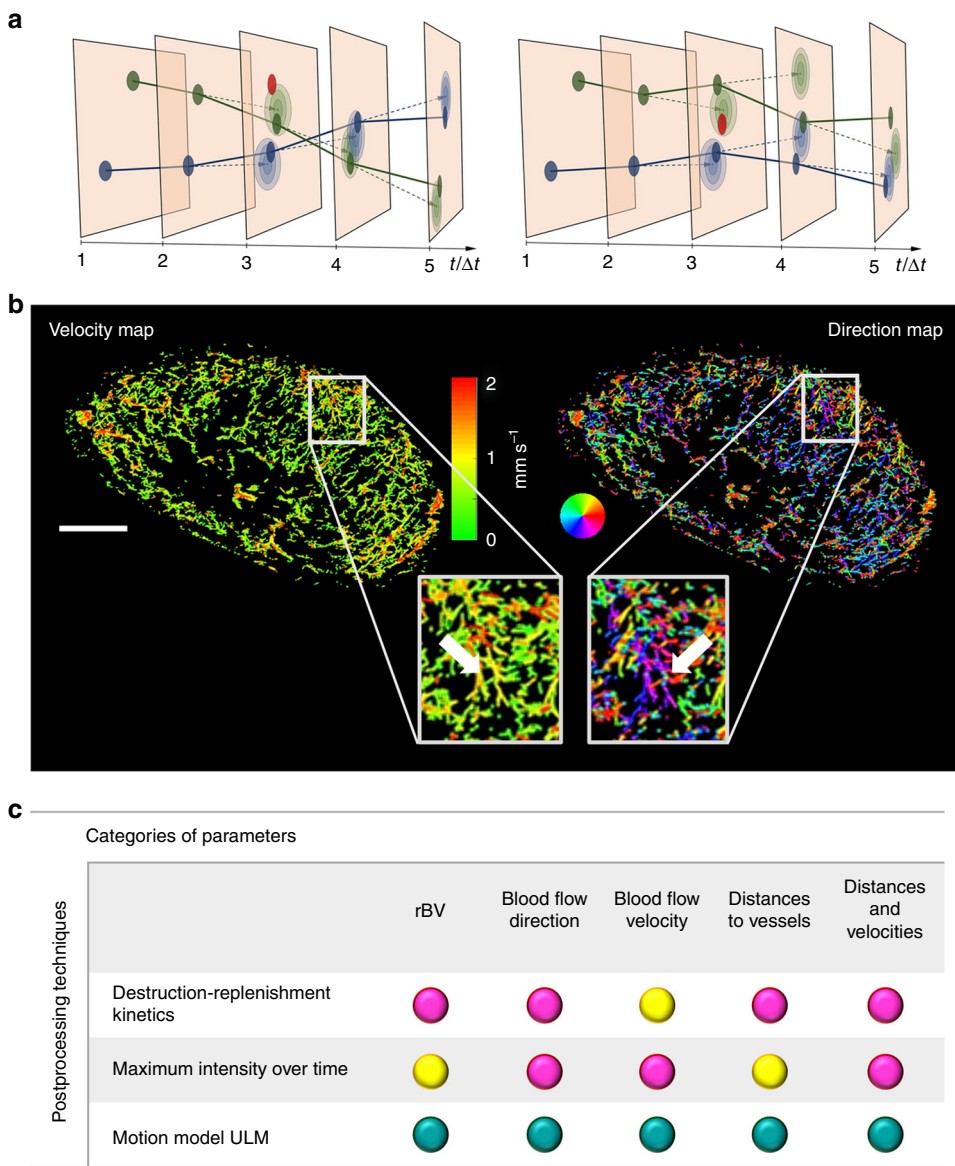

**Fig. 1** Motion model ultrasound localization microscopy (mULM): Principle, examples, and assessable parameters. **a** Sketch illustrating the principle of mULM. The filled circles mark the positions of detected MB. The red circles indicate detected MB supposed to be false alarms. The colors (blue/green) indicate the association of the MB to different tracks. One possible association of MB tracks is shown in the left diagram, another one in the right. The lighter ellipses indicate the probability density functions for the positions predicted by a linear motion model. From these, the likelihoods of the detected positions for an association are determined. The Markov Chain Monte Carlo Data Association (MCMCDA) algorithm searches for the association that maximizes the posterior probability. This also accounts for prior probabilities, like, e.g., the probability of false alarms. Taking these analyses into account, in this example, the left association is more probable than the right one. **b** Super-resolution ultrasound images of an A431 tumor provide detailed information on the microvascular architecture including insights into vascular connectivity and the number of vascular branching points (see arrows in magnifications). Functional information such as MB velocities (left image) and MB flow directions (right image; color-coding illustrating the direction of flow according to the colored circle) can be determined for each individual vessel and evaluated together with the morphological characteristics. Scale bar = 1 mm. **c** Overview of the parameter classes obtained with mULM and their accessibility with standard contrast-enhanced ultrasound methods (turquoise dot: quantitative and robust assessment of a parameter is possible; yellow dot: the information is available but its assessment is less robust, less accurate, or not quantitative; magenta dot: the parameter cannot be obtained with the respective method)

In good agreement with their histological vascular phenotypes, A431 tumors had the highest rBV values, followed by MLS and A549 tumors (Fig. 3a). However, due to the high heterogeneity of rBV within the respective tumor models, our group sizes were not sufficiently large to generate significant differences.

As expected from the visual inspection of the parametric maps (Fig. 2c), the local flow direction entropy values, describing the order of blood direction profiles, were lower in A431 than in MLS and A549 tumors but showed too large variances to separate between tumor models (Fig. 3b). Additionally, we used mULM to determine the inflow and outflow of vessels with high and low flow velocities that would correspond to arteries or veins. We found that this hierarchy, which is normally present in healthy tissues, is disturbed in tumors, where the majority of vessels are angiogenic with many arterio-veneous shunts and chaotic directions (Supplementary Figure 2 and Supplementary Table 1).

Surprisingly, parameters related to MB velocity were very similar across the tumor models, indicating that, despite their

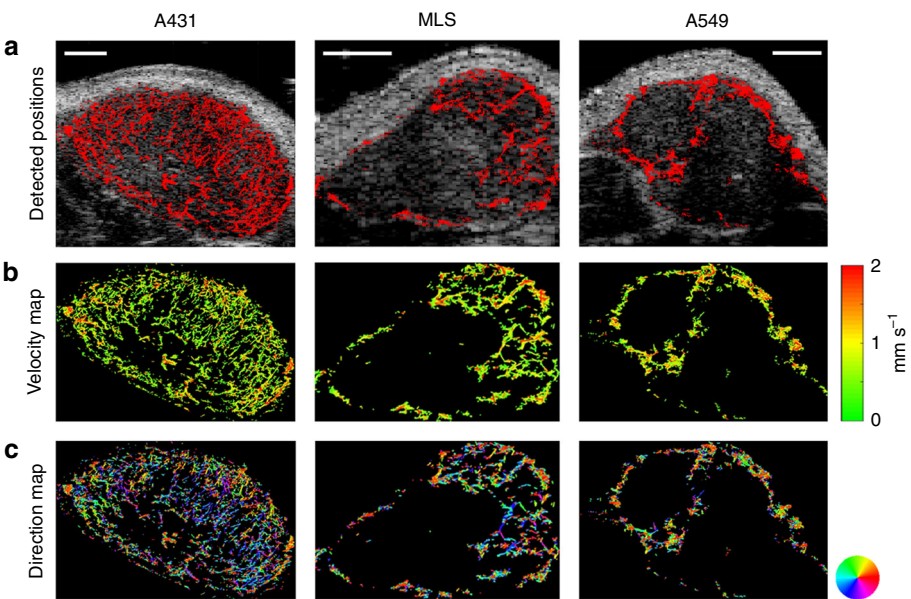

**Fig. 2** mULM-based parameter maps of tumors with different vascular phenotypes. The color-coded maps indicate **a** the detected positions of MB overlaid on the B-mode images, representing the relative blood volume, **b** individual MB velocities, and **c** directions of MB flow. Scale bars = 1 mm. The three tumor models can be distinguished based on their different vascular patterns and quantitative textural analysis can be performed based on the super-resolution parameter maps

different angiogenic phenotypes, these tumors tend to preserve a similar flow pattern (Fig. 3c).

The textural parameters of the tumor vascularization showed a significantly higher discriminatory potential (Fig. 3d–f). In this context, super-resolution images obtained by mULM enabled us to determine the distances to the closest vessel and to evaluate their mean, variance, maximum, and median values. Strikingly, the first three of the above parameters had the power to discriminate all three tumor groups. However, among all distance parameters, the maximum of distances to the closest vessel was one of the best performing ones, which precisely discriminated all three tumor models (A431 vs. MLS, $p < 0.01$; A431 vs. A549, $p < 0.01$ and A549 vs. MLS, $p < 0.01$) (Fig. 3d).

Two new parameter classes were introduced, which combine textural and functional information, i.e., (1) distances to vessel with low velocities and (2) distances to vessel with high velocities. While based on the parameters associated with distances to vessel with high velocities only one or two out of three possible combinations revealed significant differences, mean and maximum values of distances to vessel with low velocities differed significantly between all tumor models (Fig. 3e, f). Thus, the latter parameters were considered in the further analysis.

**Statistics for novel parameters obtained by mULM.** Parameters that could distinguish all three tumor models by statistical evaluation with one-way ANOVA and Bonferroni post-hoc tests were used for further analyses. These parameters were: (1) mean, (2) variance, and (3) maximum of distances to the closest vessel as well as (4) mean and (5) maximum of distances to vessel with low velocities (Fig. 4a).

In order to determine the discriminatory power of the parameters at the basis of individual tumors, for each of the selected parameters the nearest-neighbor classifier (NN) was applied in a leave-one-out-cross-validation, and confusion matrices were generated (Fig. 4b). It is clearly indicated that the maximum of distances to the closest vessel and maximum of distances to vessel with low velocities were best suitable for classifying three tumor models. With both parameters, a completely correct classification of all tumors was achieved (100%). With variance of distances to the closest vessel, 83% of the classifications were correct. However, all A431 tumors were classified correctly, and only one MLS tumor was wrongly assigned as an A549 tumor and one A549 tumor as a MLS tumor. Furthermore, with the mean of distances to the closest vessel and the mean of distances to vessel with low velocities, 67% and 58% of the correct classification could be achieved, respectively (Fig. 4b).

In order to decide which parameters should be combined to correctly classify the tumors, it is important to investigate their interdependence. The lower the correlation between parameters that have high distinctive power, the higher is the probability that they will provide complementary information. For this purpose, a correlation matrix was generated. The superior distance parameters were all highly correlated and one single out of these parameters was sufficient to distinguish all tumors. Thus, a combination of parameters was not required. Nevertheless, it may become necessary to combine parameters during examinations of animals or patients with more heterogeneous tumors. In this context, a promising parameter is the local flow direction entropy, since it shows low correlations ($r < 0.5$) with the distance and velocity parameters (Fig. 4c).

**Comparison of parameters from mULM and reference methods.** To assess the robustness and the accuracy of mULM, we firstly compared the level of tumor vascularization (rBV) obtained by mULM to rBV values obtained by three other techniques, i.e., MIOT postprocessing, ex vivo micro-computed tomography (µCT), and immunohistochemical (IHC) analysis of the tumor sections. Although rBV values did not differ significantly across the three tumor models, all modalities showed the same trend, classifying A431 tumors as the most vascularized ones, followed by MLS and A549 tumors (Fig. 5a–e). However, at a quantitative scale rBV determined by MIOT revealed higher, µCT comparable and IHC slightly lower values than by mULM (Fig. 5e).

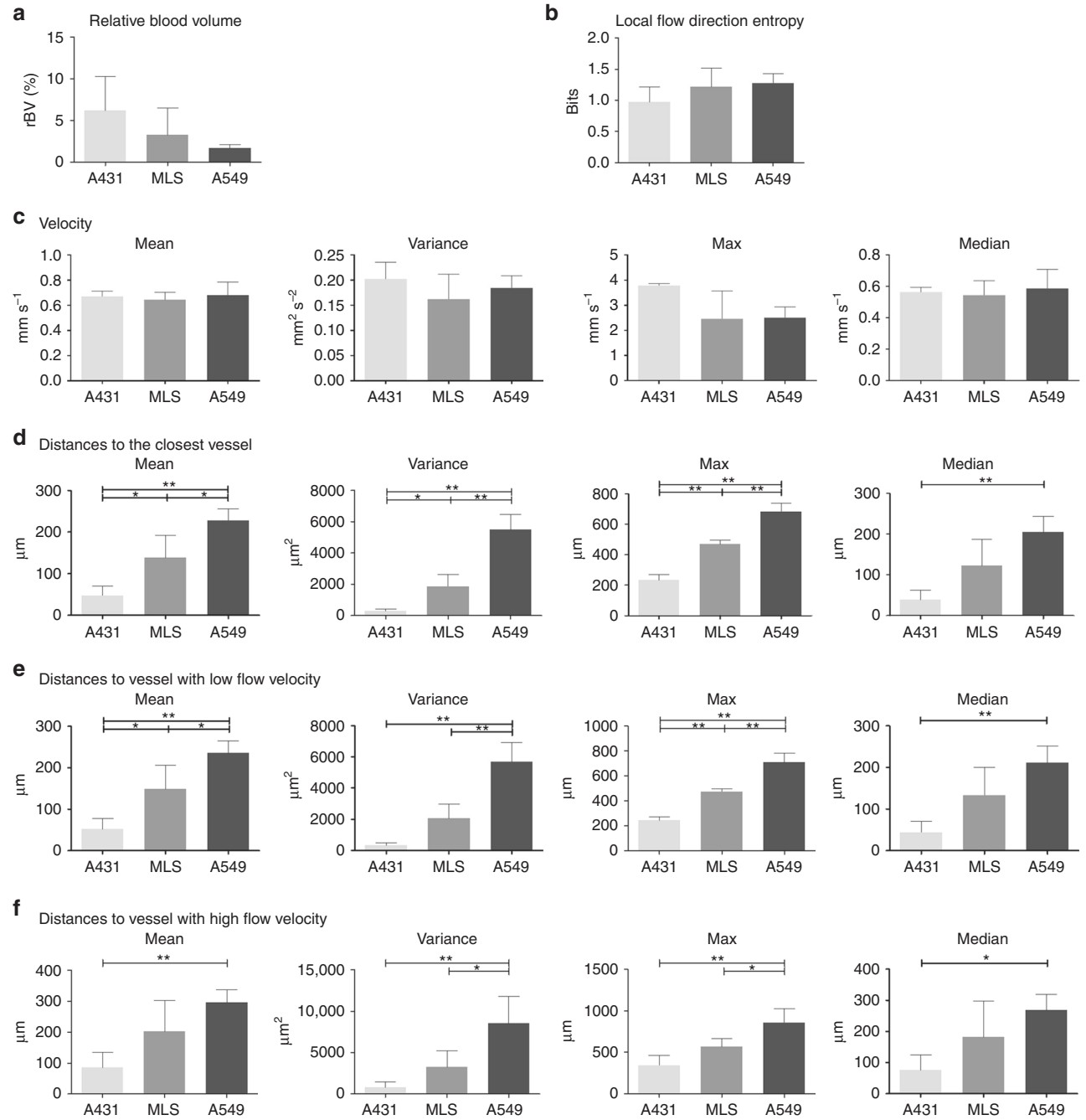

**Fig. 3** Comparison of mULM parameters. While rBV (**a**), local flow direction entropy (**b**), and MB velocities (**c**) did not differ significantly between A431, MLS, and A549 tumors, the tumor models could be distinguished using the parameters of distances to the closest vessel (**d**), and the parameters that combined velocity and distance information, i.e., distances to vessel with low (**e**) and high velocities (**f**). Only parameters that could distinguish all three tumor models were used for further analysis. For all bar plots shown, data are expressed as the mean ± s.d. ($n = 4$ per tumor model; \*\*$p < 0.01$; \*$p < 0.05$ by one-way ANOVA with Bonferroni post-hoc analysis)

In the next step, we compared mean velocities obtained by mULM with mean velocities calculated from replenishment kinetics. We found that both methods did not show differences in perfusion among the tumor models and provided values clearly below 1 mm s$^{-1}$. However, while mULM indicated mean velocities of ~0.67 mm s$^{-1}$, the values obtained by replenishment analysis were systematically lower (~0.09 mm s$^{-1}$) (Fig. 5f).

Finally, the quantitative values (mean, variance, and maximum) of distances to the closest vessel obtained by IHC und

mULM presented with the same order, i.e., A431 had the smallest, A549 the largest and MLS intermediate values (Fig. 5g).

**Clinical proof of concept**. In order to demonstrate that mULM can be performed using conventional US devices and clinically approved US contrast agents, three patients were investigated. All patients were scanned in harmonic contrast mode and B-mode using the 14L5 10 MHz transducer of the Toshiba Aplio 500 device (PLT1005BT, Canon Medical Systems, Otawara, Japan),

after slow injection of 0.5 ml of SonoVue (Bracco, Milan, Italy). The first patient was a 52-year old woman with an inflammatory HER2-positive breast cancer. mULM successfully captured a large vascular pedicle with many feeding and draining vessels at the bottom part of the tumor as well as a vascular network that was homogeneously distributed throughout the tumor tissue with higher vascularized areas at the right side of the tumor (Fig. 6a).

The second patient was a 49-year old woman with a Triple Negative Breast Cancer (TNBC) who received epirubicin/cyclophosphamide neoadjuvant chemotherapy every 3 weeks

**a** One-way ANOVA and Bonferroni post-hoc testing of parameters obtained by mULM

| | | One-way ANOVA | Bonferroni post-hoc | | |
|---|---|---|---|---|---|
| *p<0.05, **p<0.01 | | | A431/MLS | A431/ A549 | MLS/ A549 |
| Relative blood volume | | 0.15 | 0.61 | 0.19 | 1 |
| Local flow direction entropy | | 0.22 | 0.55 | 0.32 | 1 |
| Velocity | Mean | 0.76 | 1 | 1 | 1 |
| | Variance | 0.31 | 0.41 | 1 | 1 |
| | Maximum | 0.04 | 0.07 | 0.08 | 1 |
| | Median | 0.81 | 1 | 1 | 1 |
| Mean of distances to | the closest vessel | <0.01 | 0.02 | <0.01 | 0.02 |
| | vessel with low flow velocity | <0.01 | 0.02 | <0.01 | 0.04 |
| | vessel with high flow velocity | <0.01 | 0.13 | <0.01 | 0.25 |
| Variance of distances to | the closest vessel | <0.01 | 0.04 | <0.01 | <0.01 |
| | vessel with low flow velocity | <0.01 | 0.07 | <0.01 | <0.01 |
| | vessel with high flow velocity | <0.01 | 0.48 | <0.01 | 0.03 |
| Maximum of distances to | the closest vessel | <0.01 | <0.01 | <0.01 | <0.01 |
| | vessel with low flow velocity | <0.01 | <0.01 | <0.01 | <0.01 |
| | vessel with high flow velocity | <0.01 | 0.13 | <0.01 | 0.04 |
| Median of distances to | the closest vessel | <0.01 | 0.09 | <0.01 | 0.09 |
| | vessel with low flow velocity | <0.01 | 0.08 | 0.02 | 0.14 |
| | vessel with high flow velocity | 0.02 | 0.25 | 0.02 | 0.45 |

**b** Confusion matrices of mULM parameters that distinguished three tumor models

Explanatory example

Actual class: A431, MLS, A549 (columns)
Predicted class: A431, MLS, A549 (rows)

|  | A431 | MLS | A549 |
|---|---|---|---|
| A431 | X | X | X |
| MLS | X | X | X |
| A549 | X | X | X |

Distances to the closest vessel

Mean
| 3 | 1 | 0 |
| 1 | 2 | 1 |
| 0 | 1 | 3 |
67%

Max
| 4 | 0 | 0 |
| 0 | 4 | 0 |
| 0 | 0 | 4 |
100%

Variance
| 4 | 0 | 0 |
| 0 | 3 | 1 |
| 0 | 1 | 3 |
83%

Distances to vessel with low flow velocity

Mean
| 3 | 1 | 0 |
| 1 | 1 | 1 |
| 0 | 2 | 3 |
58%

Max
| 4 | 0 | 0 |
| 0 | 4 | 0 |
| 0 | 0 | 4 |
100%

**c** Correlation of mULM parameters determined by Pearson correlation coefficient (r)

| | | Relative blood volume | Local flow direction entropy | Velocity Mean | Velocity Variance | Velocity Maximum | Velocity Median | Mean of distances to the closest vessel | Mean of distances to vessel with low flow velocity | Mean of distances to vessel with high flow velocity | Variance of distances to the closest vessel | Variance of distances to vessel with low flow velocity | Variance of distances to vessel with high flow velocity | Maximum of distances to the closest vessel | Maximum of distances to vessel with low flow velocity | Maximum of distances to vessel with high flow velocity | Median of distances to the closest vessel | Median of distances to vessel with low flow velocity | Median of distances to vessel with high flow velocity |
|---|---|---|---|---|---|---|---|---|---|---|---|---|---|---|---|---|---|---|---|
| Relative blood volume | | 1.00 | 0,19 | 0.07 | 0.12 | 0.71 | 0.12 | −0.72 | −0.73 | −0.78 | −0.49 | −0.49 | −0.48 | −0.59 | −0.57 | −0.60 | −0.74 | −0.75 | −0.79 |
| Local flow direction entropy | | 0,19 | 1.00 | 0,01 | −0,44 | −0,23 | 0,04 | 0,44 | 0,42 | 0,33 | 0,49 | 0,48 | 0,37 | 0,49 | 0,49 | 0,36 | 0,41 | 0,40 | 0,30 |
| Velocities | Mean | 0.07 | 0,01 | 1.00 | 0.62 | 0.35 | 0.98 | 0.03 | 0.04 | −0.06 | 0.26 | 0.27 | 0.28 | 0.14 | 0.16 | 0.15 | 0.02 | 0.02 | −0.06 |
| | Variance | 0.12 | −0,44 | 0.62 | 1.00 | 0.72 | 0.60 | −0.35 | −0.33 | −0.38 | −0.08 | −0.05 | 0.00 | −0.21 | −0.16 | −0.17 | −0.37 | −0.36 | −0.40 |
| | Maximum | 0.71 | −0,23 | 0.35 | 0.72 | 1.00 | 0.36 | −0.73 | −0.72 | −0.75 | −0.43 | −0.40 | −0.35 | −0.59 | −0.55 | −0.53 | −0.75 | −0.74 | −0.77 |
| | Median | 0.12 | 0,04 | 0.98 | 0.60 | 0.36 | 1.00 | 0.04 | 0.04 | −0.08 | 0.30 | 0.30 | 0.31 | 0.18 | 0.20 | 0.18 | 0.02 | 0.02 | −0.10 |
| Mean of distances to | the closest vessel | −0.72 | 0,44 | 0.03 | −0.35 | −0.73 | 0.04 | 1.00 | 1.00 | 0.96 | 0.90 | 0.90 | 0.84 | 0.92 | 0.92 | 0.89 | 0.99 | 0.99 | 0.94 |
| | vessel with low flow velocity | −0.73 | 0,42 | 0.04 | −0.33 | −0.72 | 0.04 | 1.00 | 1.00 | 0.97 | 0.90 | 0.90 | 0.85 | 0.92 | 0.91 | 0.90 | 0.99 | 0.99 | 0.95 |
| | vessel with high flow velocity | −0.78 | 0,33 | −0.06 | −0.38 | −0.75 | −0.08 | 0.96 | 0.97 | 1.00 | 0.81 | 0.82 | 0.82 | 0.84 | 0.85 | 0.89 | 0.96 | 0.96 | 0.99 |
| Variance of distances to | the closest vessel | −0.49 | 0,49 | 0.26 | −0.08 | −0.43 | 0.30 | 0.90 | 0.90 | 0.81 | 1.00 | 0.99 | 0.94 | 0.95 | 0.95 | 0.92 | 0.85 | 0.85 | 0.75 |
| | vessel with low flow velocity | −0.49 | 0,48 | 0.27 | −0.05 | −0.40 | 0.30 | 0.90 | 0.90 | 0.82 | 0.99 | 1.00 | 0.96 | 0.94 | 0.96 | 0.93 | 0.84 | 0.85 | 0.76 |
| | vessel with high flow velocity | −0.48 | 0,37 | 0.28 | 0.00 | −0.35 | 0.31 | 0.84 | 0.85 | 0.82 | 0.94 | 0.96 | 1.00 | 0.88 | 0.91 | 0.95 | 0.77 | 0.78 | 0.74 |
| Maximum of distances to | the closest vessel | −0.59 | 0,49 | 0.14 | −0.21 | −0.59 | 0.18 | 0.92 | 0.92 | 0.84 | 0.95 | 0.94 | 0.88 | 1.00 | 0.99 | 0.93 | 0.86 | 0.86 | 0.77 |
| | vessel with low flow velocity | −0.57 | 0,49 | 0.16 | −0.16 | −0.55 | 0.20 | 0.92 | 0.91 | 0.85 | 0.95 | 0.96 | 0.91 | 0.99 | 1.00 | 0.94 | 0.85 | 0.85 | 0.78 |
| | vessel with high flow velocity | −0.60 | 0,36 | 0.15 | −0.17 | −0.53 | 0.18 | 0.89 | 0.90 | 0.89 | 0.92 | 0.93 | 0.95 | 0.93 | 0.94 | 1.00 | 0.83 | 0.83 | 0.81 |
| Median of distances to | the closest vessel | −0.74 | 0,41 | 0.02 | −0.37 | −0.75 | 0.02 | 0.99 | 0.99 | 0.96 | 0.85 | 0.84 | 0.77 | 0.86 | 0.85 | 0.83 | 1.00 | 1.00 | 0.96 |
| | vessel with low flow velocity | −0.75 | 0,40 | 0.02 | −0.36 | −0.74 | 0.02 | 0.99 | 0.99 | 0.96 | 0.85 | 0.85 | 0.78 | 0.86 | 0.85 | 0.83 | 1.00 | 1.00 | 0.96 |
| | vessel with high flow velocity | −0.79 | 0,30 | −0.06 | −0.40 | −0.77 | −0.10 | 0.94 | 0.95 | 0.99 | 0.75 | 0.76 | 0.74 | 0.77 | 0.78 | 0.81 | 0.96 | 0.96 | 1.00 |

(Fig. 6b). The patient was repeatedly imaged after the slow injection of 0.5 ml of SonoVue, before the start of the chemotherapy, after the first, the second and the third chemotherapy cycle. mULM super-resolution images displayed the tumor vasculature in great detail and depicted the change in blood perfusion and flow direction over the course of treatment. At the baseline measurement, the vascularization was predominantly located in the periphery of the tumor, while the center of the tumor was only modestly vascularized. Surprisingly, after the first chemotherapy cycle the tumor vascularization increased and became more homogenous with only one part on the bottom left side of the tumor remaining hypovascular. In this context, rBV measured with mULM increased from 0.04% to 1.8% while the tumor size decreased from 1.78 to 0.98 cm$^3$. After the second and third cycle of the chemotherapy the tumor size further decreased to 0.5 and 0.15 cm$^3$, respectively, while the vascularization remained enhanced. We hypothesize that tumor cell death induced by the initial chemotherapy cycles, decreased the solid stress and/or interstitial fluid pressure[21], which then caused vascular decompression and thus improved tumor perfusion and drug delivery in subsequent chemotherapy cycles. We observed the same effect in the third patient, a 56-year old woman with TNBC. At the baseline measurement, the vascularization was mainly located in the central part of the tumor without a dominant blood flow direction. After the first chemotherapy cycle, the tumor vascularization appeared much more homogeneous and strongly enhanced at the periphery. Additionally, rBV increased while the tumor size decreased from 25.6 to 3.0 cm$^3$ after the first and to 0.8 cm$^3$ after the second chemotherapy cycle, respectively (Supplementary Figure 4).

## Discussion

In this study, we evaluated mULM for the structural and functional imaging of vascular features in murine and human tumors at super-resolution. We show that mULM opens new avenues for textural and functional tumor analysis at the individual vessel level and it provides novel classes of vascular parameters. In this context, reliable and robust quantification was achieved and different (vascular) phenotypes of tumors could be accurately discriminated. We postulate that this comprehensive and quantitative vascular characterization can be clinically highly valuable since the level of vascularization, microvessel density, and the distribution of vessels are often highly correlated with tumor invasiveness, aggressiveness, metastatic potential, and prognosis of the disease[22,23], as already shown in different types of tumors, e.g., brain tumors and melanoma[24,25].

When comparing mULM to reference techniques, we found that rBV values of tumors obtained by MIOT, μCT, and IHC showed the same trend across the tumor models. Although dynamic-contrast-enhanced ultrasound (DCE-US) with parameter extraction from signal intensity time curves is current

standard for the quantification of the patients' data sets[7,9], we decided to use MIOT for the determination of the rBV since this postprocessing method showed higher robustness and accuracy in the quantification of the level of vascularization in small animals[26]. This is due to the fact that even small differences in the speed of MB injection and the animals' blood circulation under anesthesia can strongly affect the upslope and the peak of the signal intensity time curve, while it has significantly less impact on the plateau level of the MIOT curve. At a quantitative scale rBV values obtained by mULM and μCT were very similar. However, as expected, MIOT overestimated the rBV since this technique counts every US voxel showing a positive MB signal as vessel, even if the vascular fraction within the respective voxel is small. In contrary, the somewhat lower rBV values obtained by IHC are explained by the fact that we preserved samples in formaldehyde, which is known to lead to tissue shrinkage[27,28].

Subsequently, mean velocities obtained by mULM and replenishment kinetics analysis were compared. In this context, it should be noted that in 25% of the cases, replenishment curves could not be fitted due to high noise levels in the US images, while all measurements were reliably postprocessed with mULM, which demonstrates its higher robustness. Nevertheless, both methods indicated that velocity values of the three tumor models were very similar. However, the values obtained by replenishment kinetics analysis were systematically lower than by mULM. Considering velocity values of mice tumors reported in literature (1.1–1.5 mm s$^{-1}$) from multiphoton laser scanning microscopy[29], the quantitative numbers provided by mULM appear more realistic. In line with this, in previous publications the authors already reported that quantitative values obtained by replenishment kinetics analysis may not always be absolutely accurate[29–31]. This may be explained by the fact that within a region of interest blood flow within the image plane cannot be detected and therefore remains unconsidered[31]. Furthermore, the majority of replenishment analyses does not consider the influence of the beam elevation characteristics on the replenishment curve shape, which may also make the velocity values less accurate[30]. An interesting alternative reference method for blood velocities might be high-end Doppler methods, which were unfortunately not available to us. Indeed, Demené and coworkers[32] impressively demonstrated that these methods are capable to measure flow speeds down to 2.6 mm s$^{-1}$, which is sufficient to characterize the majority of physiological tissues. However, our experimental angiogenic tumors with their dysfunctional vascularization have a significant fraction of vessels with clearly lower velocity values that would be missed in the analysis and where mULM extends the measurement range to less than 1 mm s$^{-1}$.

Next, the distances obtained by mULM were compared to distances assessed from the histological sections. Even though distances obtained by μCT would yield more accurate results, since they provide 3D information, the beam-hardening artefacts hindered proper segmentation of the small blood vessels and

---

**Fig. 4** Capability of mULM parameters to distinguish tumors with different vascular phenotypes. **a** Results of the inter-group comparison of all parameters using the one-way ANOVA and Bonferroni post-hoc test. Differences between parameters with $p < 0.01$ are highlighted in dark green. Differences with $p < 0.05$ are indicated in light green. Only the parameters which could discriminate all three tumor models were used to generate confusion matrices. **b** Confusion matrices were generated to assess the capability of the parameters to correctly assign individual tumors to their according group. The numbers in the diagonal elements of the matrix represent correct classifications (highlighted in green), the remaining numbers indicate false assignments (highlighted in pink; see explanatory example in the upper row). Confusion matrices of the maximum of distances to the closest vessel and of the maximum of distances to vessel with low velocities reveal a correct classification in all cases (100%). For the variance of distances to the closest vessel, 83% correct classification is achieved. **c** Although several parameters alone already allowed a correct assignment of all tumors, parameter combinations may be required when investigating more heterogeneous tumor populations. Therefore, a correlation matrix (Pearson's correlation coefficient ($r$)) of all mULM parameters was generated to indicate the parameters providing complementary information. The highly discriminating distance parameters strongly correlated and thus, their combination may not be advantageous. However, the parameter local flow direction entropy showed a low correlation with the distance parameters and could be selected as a potential candidate for a multi-parameter readout

made the assessment of the distances unreliable. Therefore, distances to the closest vessel calculated from the histological sections were used as reference values. Although the distances calculated from histological specimen had the same trend among the tumor models as those measured by mULM, they were a bit lower. As already discussed for rBV, this can be attributed to the

fixation process of the tumor tissue and the loss of interstitial fluids during tumor removal.

There is substantial need for further improvement of the mULM technique. In this context, our first patient measurements indicated several issues that, as long as unresolved, stand in the way of a broad clinical implementation. All measurements

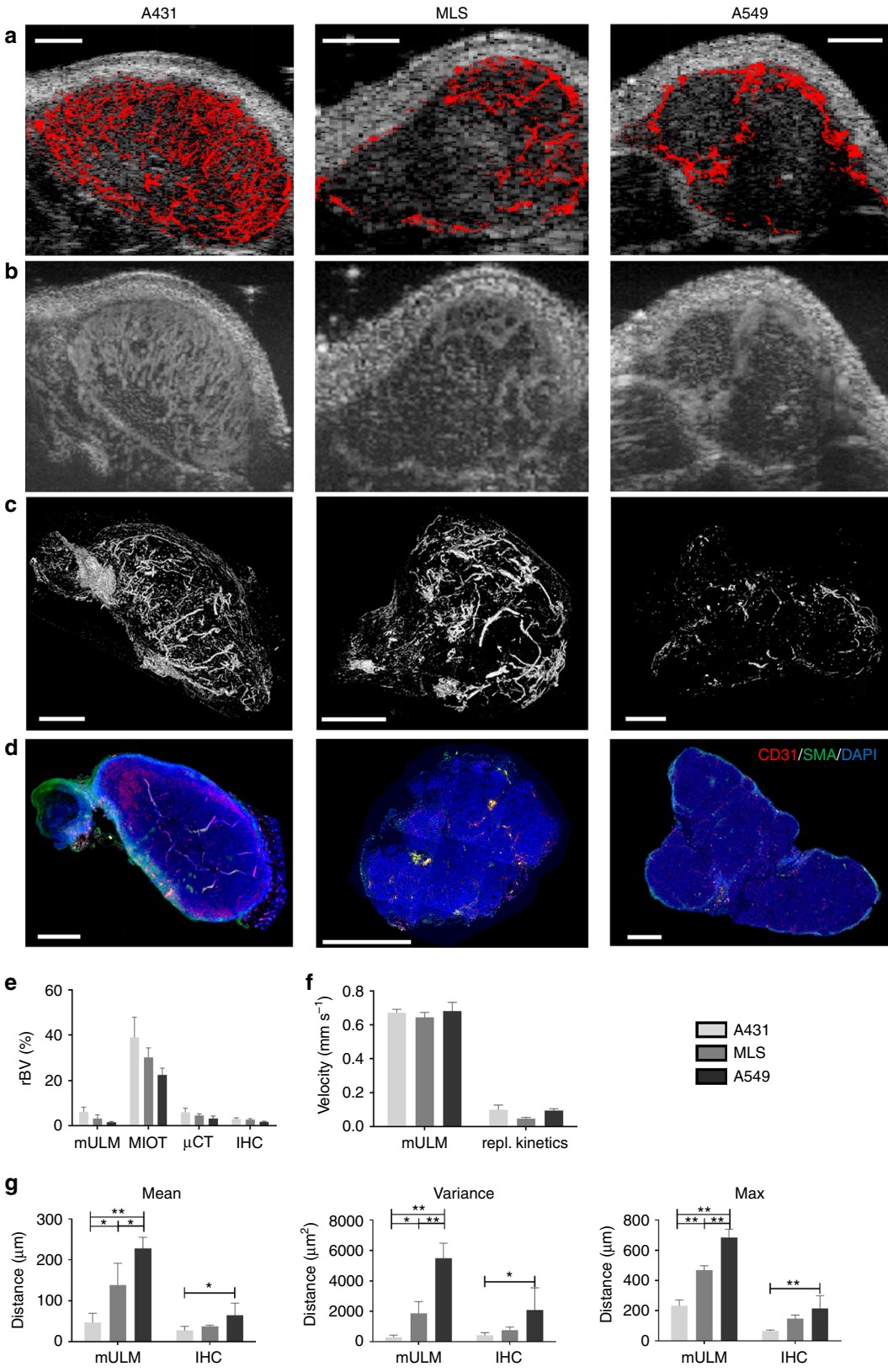

suffered from tissue motions that need to be compensated. While this is considerably easy for in-plane motion, out-of-plane movements cannot be corrected in 2D measurements, except when removing the non-matching slices, which leads to a loss of valuable data. Furthermore, the injection speed and concentration of MB need to be optimized. Therefore, we used the CEUS sequences acquired in the early phase of the injection, which reduced the number of exploitable image frames. Consequently, some vessel trees might not have been completely reconstructed. Thus, CEUS scans for mULM should be performed under slow MB injection and will require much lower MB doses than for conventional methods, which in turn, however, may help to decrease potential side effects and concerns that have been raised, e.g., for using CEUS in patients with instable cardiopulmonary conditions and pulmonary hypertension[33]. Another translational challenge represents the slice thickness of the image plane, which is larger in the clinical US scanners than in preclinical US systems, and therefore, significant overlays of vessels are expected. As mULM builds tracks based on a probabilistic choice, track continuations may also be wrong with a certain probability, which will be higher for short tracks that occur more often in ambiguous situation as in our preliminary clinical data (for a histogram of track lengths see Supplementary Figure 5). Additionally, the overlays make it difficult to correctly connect the tracks of different MB. To overcome this problem, Lin et al. detected positions of the MB at super-resolution in 3D with a stepwise motorized motion stage, a high frame-rate system and a long acquisition time[34]. Although they generated 3D super-resolution images of the vasculature, the individual MB were not tracked over time, thus the quantitative information about the hemodynamics was not obtained. We believe that the use of matrix transducers[35] for 3D mULM measurements may represent the ideal way to reconstruct and quantify the vascular network more completely and accurately.

In summary, our results demonstrate that mULM is a robust and reliable method that can be applied to data of commercial US systems. mULM can depict and accurately quantify important characteristics of the tumor vascularization at the individual vessel level and can generate new classes of vascular biomarkers that show superior performance over other CEUS methods in discriminating tumors. By providing super-resolution images of tissue vascularization, mULM offers new opportunities for a robust pattern analysis in US imaging and, in our opinion, has the potential to become an indispensable tool in tumor diagnosis and therapy monitoring. Moreover, mULM may not only be applied in oncology but may also be relevant for other indications, e.g., for characterizing inflamed tissues (e.g., inflammatory bowel disease), organ fibrosis (e.g., in liver and kidney), and immunological disorders (e.g., rheumatological disorders and organ rejection after transplantation). Furthermore, it may be applied to monitor vascular remodeling in the cardiovascular field (e.g., revascularization in ischemic tissues) and to assess antiangiogenic treatment effects, e.g., in retinopathies. Based on the presented data, we are confident

that, after further refinements, mULM will experience a rapid translation into clinical practice.

## Methods

**Study design**. The objective of this study was to establish mULM as a CEUS postprocessing method for distinguishing tumors with different vascular phenotypes at super-resolution, as well as to provide proof of principle for applying mULM on clinical data. The study consisted of three major parts. In the first part, the ability of mULM to distinguish tumor models with different vascular phenotypes was assessed[36]. For this purpose, A431, MLS, and A549 tumor xenografts were induced in female immunodeficient CD1-nude mice ($n = 4$ mice per tumor model). CEUS imaging was performed, and various parameters were extracted that describe morphological and functional vascular characteristics. Statistical tests were applied to investigate the diagnostic potential of these parameters for discriminating three tumor models and to find their ideal combination. In the second part of the study, we evaluated the robustness of mULM and the accuracy of the outcome parameters using available reference techniques. For that purpose, CEUS cine loops were analyzed with MIOT and replenishment kinetics to assess rBV values and mean velocities, respectively. For further validation of the rBV values, high resolution μCT scans of Microfil perfused tumors and histological analyses of tumor sections were evaluated ($n = 4$ mice per tumor model). In the third part of the study, we applied our technique on CEUS data from three patients, in order to demonstrate its translational potential.

**Study approval**. All animal experiments were approved by the governmental animal care and use committee (LANUV).

Clinical CEUS data derived from a study registered at clinicaltrials.gov under the number: NCT03385200. The study was approved by the RWTH Aachen University ethics committee. Written informed consent was obtained from all participants for CEUS imaging and the use of data for studying neoadjuvant chemotherapy responses.

**Cell culture**. The human cancer cell lines A431 (epidermoid carcinoma, catalog no. 300112), and A549 (non-small cell lung carcinoma, catalog no. 300114) were obtained from Cell Lines Service (Heidelberg, Germany), and the MLS cell line (ovarian carcinoma) was kindly provided by the Weizmann Institute of Science (Rehovot, Israel). A431 and MLS tumor cells were maintained in Roswell Park Memorial Institute 1640 medium (RPMI) and α-Minimum Essential Medium (α-MEM), respectively, and A549 cells were cultivated in Dulbecco's Modified Eagle Medium (DMEM). All media (Life Technologies, Darmstadt, Germany) were supplemented with 10% fetal bovine serum and 1% Penicillin/Streptomycin (Gibco, Invitrogen, Germany). Cells were incubated at 37 °C in 5% $CO_2$ and passaged at 80%–90% confluence. All cell lines were regularly checked for mycoplasma contamination.

**Xenograft tumor models**. The mice were housed in groups of four per cage under specific pathogen-free conditions with a 12 h light and dark cycle in a temperature- and humidity-controlled environment (according to FELASA-guidelines). Water and standard pellets for laboratory mice (Sniff GmbH, Soest, Germany) were offered ad libitum. Human tumor xenografts were induced in 8-weeks old female immunodeficient CD1-nude mice (Crl:CD1-$Foxn1^{nu}$, Charles River, Sulzfeld, Germany) ($n = 4$ mice per tumor model). For this purpose, $4 \times 10^6$ A431, MLS or A549 tumor cells were injected subcutaneously into the right flank. When tumors reached a size of ~5 mm × 5 mm, CEUS imaging experiments were performed. Prior to experiments, animals were anesthetized by inhalation of 2% isoflurane in oxygen.

**Contrast-enhanced ultrasound imaging**. Hard-shell polybutylcyanoacrylate (PBCA) MB were used as US contrast agent to assess the potential of mULM for tumor characterization. PBCA-MB were freshly synthetized as described before[37]. For animal experiments, the PBCA-MB suspension was diluted in sterile sodium chloride to a concentration of $2 \times 10^8$ MB ml$^{-1}$. Each mouse was injected with a 50 μL bolus containing $1 \times 10^7$ PBCA-MB over ~3 s, followed by a 20 μl saline flush, into the lateral tail vein.

**Fig. 5** Comparison of mULM parameters with reference methods. rBV values in A431, MLS, and A549 tumors were obtained by mULM (**a**), Maximum Intensity over Time (MIOT) US analysis (**b**), micro-computed tomography (μCT) (**c**), and immunohistochemistry (IHC) (**d**). Scale bars = 1 mm. All methods show a similar trend, with A431 tumors having the highest and A549 tumors the lowest level of vascularization. While MIOT clearly overestimates the rBV, μCT and mULM provide comparable values, which are in line with the data from histology (**e**). **f** Mean MB velocity values were either obtained from an exponential fit of a MB replenishment curve after a destructive US pulse from a ROI covering the entire tumor or from the mULM velocity maps. Both postprocessing procedures indicate that there are no significant differences in mean MB velocities between the tumor models. However, the mean velocity values are significantly lower in the replenishment analysis than mean velocities calculated by mULM. In **g** distance parameters determined by IHC analysis in the tumors with different vascular phenotypes are shown. Mean, variance, and maximum of the distance to the closest vessel determined by mULM had the same trend as their counterparts determined by IHC. For all bar plots shown, data are expressed as the mean ± s.d. ($n = 4$ per tumor model; *$p < 0.05$, **$p < 0.01$ by one-way ANOVA with Bonferroni post-hoc analysis)

**Image acquisition**. For the US measurements, a Vevo 2100 system equipped with the MS-550D probe (FUJIFILM Visualsonics, Toronto, ON, Canada) was used. The linear array exhibited a center frequency of 40 MHz and a bandwidth from 22 to 55 MHz. The maximum image depth was 15 mm. The focal resolution of the preclinical images is 40 μm axially, 90 μm laterally, and 200 μm elevationally.

The correct placement of the probe on the tumors was controlled prior to the measurements using real-time B-mode imaging. For the measurements, image series were recorded during the destruction-replenishment sequence at a frame rate of 50 frames per second. The images were acquired in digital raw radio frequency (RF) mode to receive uncompressed IQ data. The gain was set to 22 dB and the transmit

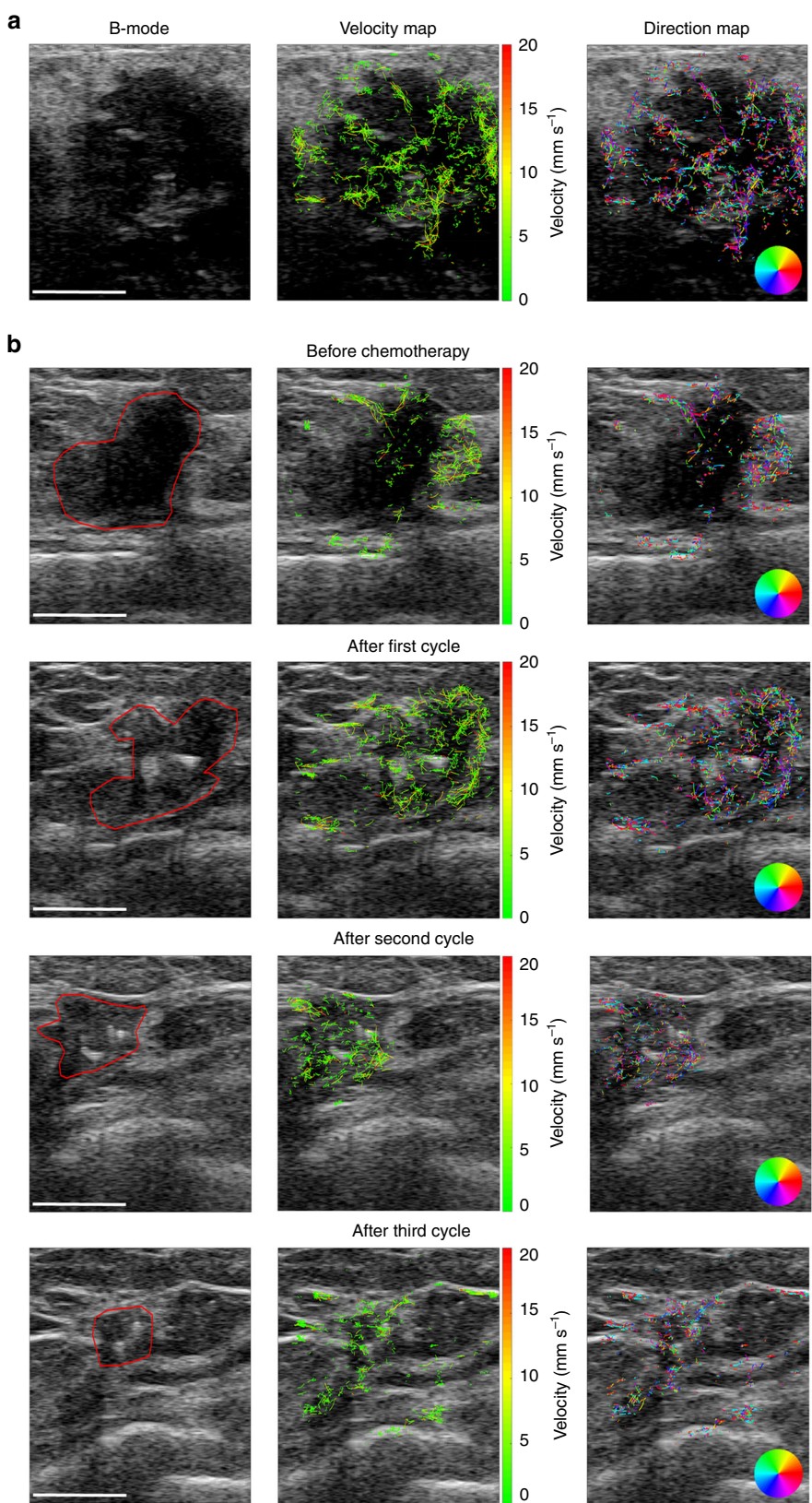

power to 2% to minimize MB destruction. For each tumor, the number of processed frames was limited to 2000, which is equivalent to 40 s measurement time.

**Image processing.** After the US measurements, the tumor border was outlined manually and the further processing steps were carried out inside the segmented area.

In a first step, a rigid motion estimation and compensation were carried out. For this, the B-mode images were interpolated on a finer grid (4-fold) to increase the accuracy of the motion compensation. The motion profiles over time typically exhibited periods of small movements disturbed by spikes of large movements due to breathing. These frames of large movements were excluded because they typically showed also strong decorrelation indicating out-of-plane movements. In the preclinical data sets, the estimated maximum displacement for all frames was $57 \pm 34$ µm, but the maximum displacement of frames which were not excluded was $21 \pm 15$ µm with a mean displacement of $10 \pm 10$ µm. Similar findings for displacement values were reported by Foiret et al.[38]. Hingot and coworkers observed displacements in the rat brain up to 40 µm in comparison to the resolution of 8 µm under optimal conditions[39]. Motion correction on these images was performed at a resolution of 1 µm, and correction accuracies below 10 µm were considered achievable with the 15 MHz imaging system.

To detect single echoes of individual MB, the B-Mode images were separated into static background and moving foreground images. The rolling background was calculated applying a temporal rank filter of rank 3 over $\pm 10$ frames around the actual frame. The foreground was computed by subtracting the background from the original frames. After applying an adaptive threshold to the foreground images, the MB were localized by calculating the intensity weighted centroid for each MB. Choosing a too low threshold leads to false detections due to noise in the image, choosing a too high threshold leads to missed detections. Therefore, for each measurement set, the threshold was adapted to result in no detections immediately after the destruction event in the destruction replenishment sequence. In the preclinical measurements, the number of detected MB positions per tumor was $4.9 \times 10^4 \pm 3.6 \times 10^4$ leading to $31 \pm 20$ MB per frame with for some tumors highly inhomogeneous spatial distribution. The comparison of the total number of detections in a given time and image area for the high-frame-rate acquisition of Errico et al.[17] ($0.17$ MB frame$^{-1}$ mm$^{-2}$) with our experiments ($2.6$ MB frame$^{-1}$ mm$^{-2}$) indicates that the detection probability for MB is comparable. Though scanning with the higher frame-rate enables to detect the same MB more often and track them continuously, it may not capture a much higher percentage of MB.

**Tracking of microbubbles.** To reconstruct the vasculature, the detected MB must be tracked over several frames. As discussed in ref. [17], usually this is either achieved by a nearest-neighbor tracking for very high frame rates in the kHz range, which need non-standard ultrafast US scanners, or by very low MB concentrations and long observation times up to several minutes.

Here, we solved the tracking challenge that occurs when using clinically recommended MB concentrations and standard US systems, by using a novel, more robust mULM method. This was necessary, since tracking by using the nearest MB in the next frame to continue a track is prone to failure under the given imaging conditions: Due to the elevational width of the imaging slice, an apparent crossing of capillaries is expected when capillaries of different directions are running in a given slice. Furthermore, the number of MB can be high (up to 100 MB per frame) for a small tumor area (e.g., ~14 mm$^2$) leading to a high MB density of >7 MB per mm$^2$. Thus, considering the applicable frame rate and the expected flow velocities, the closest MB will often not be the correct one to continue the track. Therefore, a more robust Markov Chain Monte Carlo Data Association (MCMCDA) algorithm was applied to the detected MB positions to track the MB over several frames[40]. This algorithm associates detected positions based on a probabilistic optimization considering a motion model. A detailed description of the algorithm can be found in ref. [40] but the main concepts are briefly described: Bubble positions can be associated in different ways to the tracks, as illustrated in Fig. 1a by two different track associations $\omega_1$ and $\omega_2$. The algorithm evaluates the a-posteriori probability $P(\omega|Y)$ of an association $\omega$ under the given measured positions $Y$. By Bayes' rule, this probability is proportional to

$$P(\omega|Y) \propto P(Y|\omega)P(\omega) \qquad (1)$$

In this, $P(Y|\omega)$ is the likelihood of the measured position $Y$ under a given track

association as determined from a linear motion model. The a priori probability of a track association, without the knowledge of the measurement data is $P(\omega)$, which is calculated from the assumed probabilities of false detections, missing detections, track starts, and track ends. These parameters of the algorithm are chosen as described in ref. [40].

Trying all possible track combinations and finding the association $\omega_{max}$ that maximizes the a posteriori probability: $\omega_{max} = \arg\max_{\omega \in \Omega} P(\omega|Y)$ is, however, an intractable combinatorial problem. Thus, in a Monte Carlo approach a Markov chain was used to randomly draw associations with this probability distribution. By this, associations with high probability are drawn more often and the best association will be kept. For example, in Fig. 1a, the track association $\omega_1$ had a higher probability compared to the nearest-neighbor association represented as $\omega_2$, because the position prediction of the motion model resulted in higher likelihoods $P(Y|\omega)$ of the measured positions under the assumption of track association $\omega$, which also led to a higher a posteriori probability $P(\omega|Y)$.

The tracking algorithm yielded not only the trajectories but also the velocities and directions of the moving MB. The maximum number of successive detections for one trajectory was 42 detections and the typical number is between two and ten detections.

**Definition and extraction of parameters.** For further evaluations, images of MB tracks, velocities, and flow direction were reconstructed with a pixel size of 5 µm × 5 µm for each tumor. We chose 5 µm as the pixel size, which is ~$\lambda/8$ or an eighth of the 40 µm axial extent of the point spread function of the Vevo 2100 system in the focus, based on the results of other works using bubble localization: Errico et al.[17] reported an in vivo resolution of $\lambda/6$. Viessmann et al.[41] showed that features 5.1–2.2 times smaller than the point spread function could be resolved by MB localization with a clinical system in an in vitro setup.

Desailly et al.[42] discussed the lower limits of localization accuracy as estimated from the Cramér-Rao-Lower-Bound. Using their formula results in a lower bound of localization accuracies for the Vevo MS550D transducer of 79 nm axially and 262 nm laterally even for a low signal-to-noise ratio (SNR) of 1.0 and a correlation of 1.0 (center frequency 40 MHz, bandwidth 33 MHz). However, these theoretical limits do not account for several effects that will be present for in vivo data: Inhomogeneities in the tissue's material parameters lead to additional time of flight fluctuations that can be larger than the standard deviation of time of flight measurements as considered by Desailly and coworkers[42]. Additionally, the theoretical limit assumes that MB signals are perfectly isolated with complete removal of the stationary background. Even with good motion compensation techniques, part of the background will still be present leading to speckle disturbing the exact localization of correlation maxima. Finally, the correct superposition of motion corrected MB detections is limited by the accuracy of rigid motion compensation that will not be correct for the complete image region. Based on these considerations the resolution was not expected to exceed the 5 µm pixel resolution chosen.

The measurement of the actual resolution without ground truth is difficult. As tracks are mapped with a thickness of the 5 µm pixel resolution, a vessel that is not sampled more than once will seem to be resolved at 5 µm. To get an indication of the resolution of the resulting vessel images, the number of MB passages through a pixel were counted and lateral profiles of small vessels that were passed by at least five MB were retrieved. The full-width-half-maximum of the smallest vessels was found to be ca. 10 µm as shown in Supplementary Figure 1.

MB track images were generated using Bresenham's line algorithm to connect the MB positions of the estimated tracks. The MB track image was a binary image indicating the pixels which were passed by the MB. In the flow velocity map and the flow direction map, the corresponding quantity was assigned to the pixels along the track.

For the evaluation of the parameters, each tumor was divided into two regions: a rim of 0.5 mm thickness and the core which was the remaining area when excluding the rim from the whole tumor area. For the characterization of the tumor vasculature, we used only the core region to exclude the large feeding vessels in the rim.

From the MB track image the rBV was derived as the ratio of the area covered by the tracks to the respective total area, which was expected to be proportional to the rBV.

---

**Fig. 6** Preliminary mULM results from breast cancer patients. CEUS measurements were performed with a conventional US device and phospholipid MB. Scale bars = 10 mm. In **a** B-mode and mULM images of the patient with the HER2 positive breast carcinoma are shown. mULM visualizes in detail the tumor vascular pedicle on the bottom side of the tumor that branches into smaller vessels which are distributed heterogeneously throughout the tumor, slightly denser on the right-hand side. In **b** B-mode and mULM images of a triple negative breast carcinoma in a patient treated with neoadjuvant chemotherapy are presented. Measurement were performed before (first row), and after the first (second row), the second (third row), and the third cycle of chemotherapy (fourth row). The first column shows B-mode images with the tumor borders highlighted by the red polygon, the second column displays the mULM velocity maps and the third column indicates the mULM direction maps. At the baseline measurement, the tumor vascularization was present mainly in the peripheral areas, and only modestly in the tumor core without showing any dominant direction. After the first cycle of treatment, the tumor shrank and vascularization appeared more homogeneous with only one avascular part at the bottom side of the tumor. After the second and third cycle of treatment, the tumor volume further decreased, while the level of vascularization remained stable

Additionally, from the MB track image a track distance map was generated applying the Euclidean distance transform (bwdist function, Matlab, MathWorks, Natick, MA, USA). For each pixel, the track distance map provided the shortest distance to the next track, which was used as an estimate for the distance to the closest vessel. For each track distance map, mean, variance, maximum, and median of the distances to the closest vessel were calculated for the respective areas. Small distances are characteristic for a fine meshwork of vessels. The larger the maximum distance, the larger are the non-perfused areas.

From the flow velocity map the statistics of the velocities were derived. Again, mean, variance, maximum, and median were calculated.

Since we were interested in the structure of the vasculature, we divided the vessels into two groups of high and low flow velocities, respectively, and calculated the distance parameters separately for the resulting two groups. We defined the mean value of the median velocities of the tumors as the threshold between high and low flow velocities (0.7 mm s$^{-1}$) and calculated the mean, variance, maximum, and median of the distances to vessel with low and high velocities.

We were also interested in parameterizing the directions of MB flow. To characterize a locally ordered flow with predominant directions in contrast to locally chaotic flow directions, we defined sub-regions of $25 \, \mu m \times 25 \, \mu m$ and calculated the local flow direction entropy of the vessels from the flow direction maps. Predominant directions within sub-regions will result in low entropy values. Local entropy values are averaged over the tumor area.

However, it should be noted that all CEUS data are acquired with 1D array transducers and therefore, all distance and velocity parameters are calculated in the imaging plane.

**Statistical analysis**. Data are presented as mean ± standard deviation (s.d.). Sample sizes were chosen according to our experience from pevious studies and considering the ethical demands to keep the animal number as low as possible. No randomization or blinding was performed to allocate the subjects in the study. The one-way ANOVA and Bonferroni post-hoc test were applied to evaluate differences between groups considering a $p$-value of <0.05 to be significant. The assumptions of the one-way ANOVA, which are equal variances within groups and normality of the data, were positively tested with significance level $p = 0.01$ with the Brown-Forsythe and the Shapiro–Wilks test, respectively. All analyses were performed using GraphPad Prism 5.0 (GraphPad Software, San Diego, CA, USA) and Matlab 2017b (Matlab, MathWorks, Natick, MA, USA).

**Confusion and correlation matrices**. For all parameters extracted by mULM, one-way ANOVA and Bonferroni post-hoc tests were applied. Parameters that distinguish all three tumor models were used for the further analyses. Then, for each parameter the nearest-neighbor classifier (NN) was applied in an exhaustive leave-one-out-cross-validation. The results are presented in confusion matrices, which plot the actual classes versus the classes predicted by the classification. The numbers in the diagonal elements of the matrix represent the correct classifications; the remaining numbers indicate the false assignments. The correct classification is expressed in percentage.

Additionally, a correlation matrix of all parameters obtained by mULM was generated to depict the pairwise dependencies among them measured by the Pearson's correlation coefficient ($r$).

Confusion and correlation matrices were generated by Matlab, MathWorks, Natick, MA, USA.

**Reference methods**. We firstly validated rBV obtained by mULM by rBV obtained by MIOT analysis, ex vivo μCT and histological evaluation of the tumor sections. For generation of the MIOT images, the highest amplitude values of each pixel were preserved throughout the recorded B-mode image sequence after MB injection. Then, from each CEUS image a corresponding background image, containing the median pixel-wise value of all B-mode images, was subtracted to reveal the vascular network. To assess the rBV from MIOT images, a threshold-based segmentation of the vessels was performed and rBV was calculated as the fraction of vessels in the tumor area using the Imalytics Preclinical software (Gremse-IT, Aachen, Germany)[43].

Next, in terminal experiments, mice were perfused intracardially with the silicone rubber radiopaque compound Microfil® (FlowTech, Carver, MA), which polymerizes in blood vessels within 20 min[36]. After Microfil perfusion, tumors were excised, preserved in 4% formalin, and scanned in the high-resolution desktop X-ray micro-CT system SkyScan 1172 with a Hamamatsu 10 Mp camera (pixel size 11.66 μm) (SkyScan, Kontich, Belgium). Tumors were scanned around the vertical axis with rotation steps of 0.3° at 59 kV and a source current of 167 μA. 640 projections (2096 × 4000 pixels) were acquired during 2.5–3 h per tumor. After threshold-based segmentation, rBV was determined as fraction of Microfil-perfused vessels of total tumor volume using the Imalytics Preclinical software (Gremse-IT, Aachen, Germany)[43].

Finally, tumors were embedded in paraffin and cut into 5 μm thick sections. Immunostaining of endothelial cells was performed with a rat anti-mouse CD31 primary antibody (1:10, DIA-310, Dianova, Hamburg, Germany), followed by a donkey anti-rat Cy-3-labeled secondary antibody (1:500, 712-166-153, Dianova, Hamburg, Germany). Smooth muscle cells and pericytes were labeled using a biotinylated anti-α-smooth muscle actin (α-SMA) primary antibody (1:500, D12625, Progen, Heidelberg, Germany) and streptavidin-Cy-2 (1:200, 016-220-084, Dianova, Hamburg, Germany). Nuclei were counterstained with 4,6-diamidino-2-phenylindole (DAPI, 1:500, D1306, ThermoFisher Scientific, USA). Fluorescent micrographs were obtained with an Axio Imager M2 light microscope and AxioCamMRm revision 3 high-resolution camera (Carl Zeiss Microimaging, Göttingen, Germany). For each tumor a whole histopathological section was analyzed. The vessel area fraction, referring to rBV, was calculated by semi-automated detection and filling of the lumen of CD31-positive structures and dividing the resulting area by the total tumor area. Furthermore, the vessel maturity index was determined by calculating the percentage of SMA positive vessels per total number of vessels. All IHC analyses were performed using the AxioVision Rel 4.8 software (Carl Zeiss Microimaging).

Secondly, mean velocities obtained by mULM were compared to mean MB velocities calculated by destruction replenishment analysis, which is a clinically established US method. The corresponding algorithm was implemented in a custom program (Matlab, R2015a, MathWorks, Natick, USA). Cine loops were acquired with 50 frames per second using a 40 MHz transducer. After recording the MB bolus injection phase, a destructive pulse was applied for 1 s to destroy all MB in the imaged tumor slice. Then, the replenishment of circulating MB was recorded over approximately 40 s. Velocities were determined by fitting the slope of the replenishment curve as described by Wei et al.[13].

Finally, we validated the distances to the closest vessel obtained by mULM to their counterparts obtained by IHC analysis. The distances to the closest vessel were calculated manually from five micrographs of each histopathological section, as the shortest distance from one CD31 positive to the next CD31 positive vessel wall. Then we calculated mean, variance, and maximum values. IHC analyses were performed using the AxioVision Rel 4.8 software (Carl Zeiss Microimaging).

**Patient examinations**. The patients were scanned at the Department of Gynecology and Obstetrics at the University Medical Center of the RWTH University in Aachen, Germany with a 10 MHz PLT 1005BT transducer connected to a Toshiba Aplio 500. Mechanical index that was used during examinations was 0.07 and thermal index was below 0.4. The focal resolution of the clinical images in contrast mode was determined from the full width half maximum of single microbubble point spread functions at focal depth as 560 μm axially and 780 μm laterally. The elevational resolution of 1 mm at the elevational focus in 20 mm depth was communicated by Canon Medical Systems.

In these clinical measurements the maximal displacement of the frames that were not excluded was 147 ± 26 μm, with a mean displacement of 114 ± 34 μm. The number of detected MB positions per patient was $1.8 \times 10^4 \pm 0.9 \times 10^4$, while the number of detected MB per frame was 36 ± 18. All further details about the scanning procedure are presented in the results section.

**Data availability**. The data that support the findings of this study are available within the article and supplementary files, or available from the authors upon request.

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

## Acknowledgements

The authors thank M. Weiler for performing µCT measurements and S. von Stillfried for providing paraffin embedded samples. This study is supported by Deutsche Forschungsgemeinschaft DFG (KI1072/11-1, SCHM1171-4/1).

## Author contributions

F.K. and G.S. planned the study, supervised the experiments and the data analysis, and revised the manuscript. T.O. and A.R. performed the experiments. S.D., D.A., M.P., and G.S. established the algorithm and performed the postprocessing analyses. E.S. and S.Del. provided the clinical data. B.T. and A.R. assisted with the data analyses and reviewed the manuscript. T.L. interpreted the results and revised the manuscript. T.O., S.D., and M.P. prepared the figures. T.O., and S.D. drafted the manuscript.

## Additional information

**Competing interests:** The authors declare no competing interests.

