## [Peer Review File · Nature Communications]

Reviewers' comments:

Reviewer #1 (Remarks to the Author):

The manuscript "Super-Resolution Ultrasound Bubble Tracking for Preclinical and Clinical Multiparametric Tumor Characterization" by Opacic et al is a very interesting work in the recent field of superresolution ultrasound.

The paper is clearly written and the results are nicely explained. Both preclinical and preliminary clinical results are presented.

In my opinion, the preclinical part and study of different parameters extracted from the microbubbles localization maps for vascular phenotype comparison is very interesting and well addressed and provides new insights on the future capability of Ultrasound Localization Microscopy to propose a better cancer diagnosis and cancer classification.

Nevertheless, instead of improving it, the overall quality of the manuscript is penalized by the clinical results which are definitely not convincing. The clinical image quality of the vascular network in patient breast and thyroid is really low and does not permit to be confident with the clinical results. It leads to strong overstatements in the manuscript (for example last sentence of the abstract). Personally, I would cancel the clinical results that strongly degrade the overall quality of the manuscript. The clinical configuration with much larger tissue motion artifacts remains a bottleneck for the current approach. The number of detected bubbles is insufficient to draw a sufficient part of the vascular network. The preclinical results are sufficient and deserve publication.

Several General comments should be taken into account :

It is difficult to understand why the authors introduced a new terminology Ultrasound Bubble Tracking (UBT) as its meaning is very large and broader than the proposed tracking algorithm. Authors propose an elegant and interesting tracking method within the field of Ultrasonic Bubble Tracking. Other ultrasonic bubble tracking methods were already proposed in the former articles of the field. The concept based on the ultrasonic tracking and localization of individual microbubbles was introduced as Ultrasound Localization Microscopy by analogy to the field of Superresolution Optics which is commonly named Optical Localization Microscopy. Introducing a new name for similar approaches is introducing confusion in the community and does not add new scientific information. Moreover the proposed name for the technique embraces a much broader field than the technique itself.

Abstract:

- sentence "showed superior performance in discriminating tumors with different vascular phenotypes » is overclaiming. A comparison with high end blood flow Doppler imaging methods without contrast agents should be provided.
- Last sentence is strongly overclaiming with respect to the very low quality of clinical images.

Introduction:

Line 64. Very high frame rates interest is not only to improve detection and tracking but also to acquire much more bubbles locations in a reasonable time and finally to correct for tissue motion artifacts in a robust way. This should be explained as the quality of the final superresolved image is strongly related to : the number of microbubbles correctly detected, the correction of motion artifacts strongly linked to the total acquisition time.

Results :

- Resolution was found to be 5 μm . Does it fit with the proposed theoretical analysis by Desailly et al ? Could be interesting to compare and refer
- Important parameters are missing : the average number of bubbles detected per image, the total number of bubbles detected.

- Line 368, provide more information about the average and maximal displacement estimated.
- Line 371, Large movements if not compensated do not permit to recover the exact location of the vasculature before motion. This point is key for clinical investigation. The very large difference of image quality between preclinical and clinical results comes from these much more complex configurations. Again, the number of bubbles detected by the algorithm is preclinical and clinical configuration should be given. Mapping the vasculature of tumors or organs requires a very large number of detected bubbles. These parameters should be added and discussed
-
- Figure 6 and figure 7 are really not convincing. It is impossible to argue on figure 7 that the decrease of number of detected vessels is due to the chemotherapeutic treatment. The detected bubbles trajectories do not provide a connected vascular network.
- Line 525 provide safety regulations parameters and ethical approval authorization number for the clinical study.

Reviewer #2 (Remarks to the Author):

Very interesting paper developping a new and elegant analysis of vascular heterogeneity of tumors in mice and patients with a tentative of proof of concept

But 3 points to stress:

:

1) the references need to be adapted : more recent and recognized papers as validation of DCE-US

2) why use MIOT and not DCE-US which is recognize as standard?

2) The 3 clinical exams are very heterogeneous in term of machins and settings in order to give some proof of concept

Modifications to perform :

For the references please change :

a) number 9 by
maging biomarker roadmap for cancer studies.

O'Connor JP, Aboagye EO, Adams JE, Aerts HJ, Barrington SF, Beer AJ, Boellaard R, Bohndiek SE, Brady M, Brown G, Buckley DL, Chenevert TL, Clarke LP, Collette S, Cook GJ, deSouza NM, Dickson JC, Dive C, Evelhoch JL, Faivre-Finn C, Gallagher FA, Gilbert FJ, Gillies RJ, Goh V, Griffiths JR, Groves AM, Halligan S, Harris AL, Hawkes DJ, Hoekstra OS, Huang EP, Hutton BF, Jackson EF, Jayson GC, Jones A, Koh DM, Lacombe D, Lambin P, Lassau N, Leach MO, Lee TY, Leen EL, Lewis JS, Liu Y, Lythgoe MF, Manoharan P, Maxwell RJ, Miles KA, Morgan B, Morris S, Ng T, Padhani AR, Parker GJ, Partridge M, Pathak AP, Peet AC, Punwani S, Reynolds AR, Robinson SP, Shankar LK, Sharma RA, Soloviev D, Stroobants S, Sullivan DC, Taylor SA, Tofts PS, Tozer GM, van Herk M, Walker-Samuel S, Wason J, Williams KJ, Workman P, Yankeelov TE, Brindle KM, McShane LM, Jackson A, Waterton JC.

Nat Rev Clin Oncol. 2017 Mar;14(3):169-186

b) number 7 by :

Validation of dynamic contrast-enhanced ultrasound in predicting outcomes of antiangiogenic therapy for solid tumors: the French multicenter support for innovative and expensive techniques study.

Lassau N, Bonastre J, Kind M, Vilgrain V, Lacroix J, Cuinet M, Taieb S, Aziza R, Sarran A, Labbe-Devilliers C, Gallix B, Lucidarme O, Ptak Y, Rocher L, Caquot LM, Chagnon S, Marion D, Luciani A, Feutray S, Uzan-Augui J, Coiffier B, Benastou B, Koscielny S

c) please add this ref just at the end of sentence line 44

Dynamic contrast-enhanced ultrasound parametric maps to evaluate intratumoral vascularization.

Pitre-Champagnat S, Leguerney I, Bosq J, Peronneau P, Kiessling F, Calmels L, Coulot J, Lassau N.

Invest Radiol. 2015 Apr;50(4):212-7

for the clinical part: the firsts clinical case need to be delete because no contrast specific mode was use, , 3 ml of sonovue ;WHY? this dose is not recommended images are not demonstrative

The two others case use different machine, dose of sonovue, contrast mode etc ...

Reviewer #3 (Remarks to the Author):

Tracking of individual microbubbles (ultrasound contrast agent) confined to the vasculature is used to probe the characteristics of the microvascular environment in tumors. The tracking allows to estimate morphological (distance between vessels) as well as functional (velocity and direction of blood flow) information which cannot be obtained using standard ultrasound techniques due to the diffraction limit of the imaging system. Using a preclinical ultrasound system on a mouse model, the authors shows that different tumor types can be discriminated based on information mostly related to the distance between microvessels. Preliminary data obtained during clinical exams are also presented. Overall, this work appears technically sound with novel preclinical and clinical findings, but I would like the authors to perform appropriate revisions and additions.

Although the term 'super-resolution' is employed throughout the manuscript, I am not sure the technique described here should employ it as simply determining the position of an object to an accuracy greater than the resolution limit for an imaging system is not super-resolution (the imaging bandwidth is not extended beyond the conventional limit). In the data presented, individual MBs are localized and tracked to a resolution that is finer than the imaging resolution but the 50 Hz (or lower) frame rate and recording time <1 min excludes a reconstruction depicting the actual vessel size but rather the distance between vessels and blood velocity in the micro vessels. In contrast with refs 16, 17 and 30, the number of detected MBs does not seem to provide a full map of the vasculature giving an estimation of the actual vessel sizes.

The authors claim to achieve an in-plane resolution of 5 μm ("A spatial resolution of approximately 5 μm was achieved", p.5), the manuscript needs to bring more evidence to support the 5 μm resolution obtained, especially with respect to physiological motions over the course of the recording. What displacement value is considered small or large in the mouse experiments? How

much displacement did you measure in the clinical data? Recent articles have emphasized the issue of tissue motion for single MB tracking methods (Hingot et al, *Ultrasonics*, 2017; Foiret et al., *Scientific Reports*, 2017), thus it would be interesting to include these papers in the manuscript as it seems motion will be a challenge for the clinical application of the technique.

Please provide an estimate of the voxel size for all ultrasound systems used in this work (axial, lateral and elevational resolution) as well as an estimate of the improvement achieved in plane by UBT in the results presented. As 1D arrays are used in this work, it should appear in the text that all distances and velocities are in plane and therefore not absolute as they do not consider the uncertainty in elevation.

The technique is appealing but I am worried about the physical limitations using 1D arrays. Here, a high-frequency preclinical system dedicated to small animals is used which offers a relatively narrow elevation resolution. However, with clinical systems, as stated the elevation resolution is in the mm range which, combined with motion, may prevent easy analysis. Looking at the clinical data presented, I feel that this is the case (for instance with Fig 6 a, and Fig 7) where MB tracks seem to overlap a lot. The analysis of the UBT parameters on the mouse tumors seem to indicate that the parameters related to the distance between vessels are the most discriminatory, and therefore the translation to human may greatly suffer from the 1D array limitations. If it is the case and 3D imaging is necessary, it mitigates a rapid translation toward clinic as 3D US systems are still sparse.

Please consider adding scalebars in Fig 1, 2, 5, 6 and 7 to get a better sense of the tumor dimensions and the MB tracks.

In the manuscript, you analyze the distance between vessels but the output the UBT process are individual MB tracks. Does this mean you first group MB tracks that are adjacent to one another and associate them to a vessel? If so, how do you do the association? Do you differentiate the adjacent tracks with the flow direction (for instance adjacent tracks with opposite flow direction)? Do you use the average velocity for each trajectory? What is the typical number of successive detections for a trajectory?

Regarding Fig 5g, I feel that the comparison between UBT and IHC should be between UBT and μ CT which gives the true 3D distance between vessels in the entire tumor. The imaging plane has an elevation resolution of a few hundred μ m (I am guessing) whereas IHC represents a slice of only 5 μ m, so in a way it is expected to see discrepancies between the two methods. To my eyes, Fig 5a is closer to Fig 5c than Fig 5d and I think a comparison between UBT and μ CT will yield better results.

Does the separation between high and low velocities in the analysis of the distances means you predominantly look at the arterial flow for the high velocities and venous flow for the low velocities (my knowledge of tumor vascularization is limited)? Or large versus small vessels? It would be interesting to see separated track maps for the low velocities and for the high velocities.

Reviewer 1

1. The manuscript "Super-Resolution Ultrasound Bubble Tracking for Preclinical and Clinical Multiparametric Tumor Characterization" by Opacic et al is a very interesting work in the recent field of superresolution ultrasound. The paper is clearly written and the results are nicely explained. Both preclinical and preliminary clinical results are presented. In my opinion, the preclinical part and study of different parameters extracted from the microbubbles localization maps for vascular phenotype comparison is very interesting and well addressed and provides new insights on the future capability of Ultrasound Localization Microscopy to propose a better cancer diagnosis and cancer classification.

- *We highly appreciate and thank to the reviewer for the positive comments on our work.*

2. Nevertheless, instead of improving it, the overall quality of the manuscript is penalized by the clinical results which are definitely not convincing. The clinical image quality of the vascular network in patient breast and thyroid is really low and does not permit to be confident with the clinical results. It leads to strong overstatements in the manuscript (for example last sentence of the abstract). Personally, I would cancel the clinical results that strongly degrade the overall quality of the manuscript. The clinical configuration with much larger tissue motion artifacts remains a bottleneck for the current approach. The number of detected bubbles is insufficient to draw a sufficient part of the vascular network. The preclinical results are sufficient and deserve publication.

- *We agree with the reviewer that our initial clinical data may not have looked convincing. The purpose of presenting initial clinical data is to demonstrate that bubble tracking in a clinical setting is possible and that the major limitations to derive mULM images in a clinical setting are the quality of motion compensation and the restriction to a relatively thick 2D plane. Because of the low number of MB detected, we agree with the reviewer that the clinical results did not sufficiently support the conclusions drawn. Therefore, we examined additional patients and are now able to present more convincing examples with more effective motion compensation and consequently more MB detections (see Fig. 6). In addition, we moved the less successful example to the supplement (see Supplementary Fig. 4) to underline the aspects that still need to be improved for further clinical translation.*

3. It is difficult to understand why the authors introduced a new terminology Ultrasound Bubble Tracking (UBT) as its meaning is very large and broader than the proposed tracking algorithm. Authors propose an elegant and interesting tracking method within the field of Ultrasonic Bubble Tracking. Other ultrasonic bubble tracking methods were already proposed in the former articles of the field. The concept based on the ultrasonic tracking and localization of individual microbubbles was introduced as Ultrasound Localization Microscopy by analogy to the field of Superresolution Optics which is

commonly named Optical Localization Microscopy. Introducing a new name for similar approaches is introducing confusion in the community and does not add new scientific information. Moreover, the proposed name for the technique embraces a much broader field than the technique itself.

- *We agree with the reviewer and reconsidered the name of our method. Errico et al. used Ultrafast Ultrasound Localization Microscopy (uULM) where they localized MB and tracked them in subsequent frames with high frame-rate-systems (500 frames/second) to reconstruct the vasculature and quantify the blood flow in rats' brain. For this, the authors used a nearest neighbor algorithm for MB tracking, which considers the closest MB to continue the track. Although we used the same principle for the localization of the MB, we used a different algorithm to track the MB and estimate their directions and velocity. In our study we applied the Markov Chain Monte Carlo Data Association (MCMCDA) algorithm which associates detected positions based on a probabilistic optimization considering a motion model. Therefore, we were able to sample the tumor vasculature within relatively short measurement times and with low frame rates. Furthermore, we found this tracking method well-suited for the assessment of the complex vascular network which is found in tumor tissue: As the unique aspect of our tracking method is the incorporation of a probabilistic motion model, we propose to use the established term Ultrasound Localization Microscopy (ULM) as suggested by the reviewer and express the motion-model-based tracking approach by using the term motion model Ultrasound Localization Microscopy (mULM).*

4. Abstract: sentence "showed superior performance in discriminating tumors with different vascular phenotypes » is overclaiming. A comparison with high end blood flow Doppler imaging methods without contrast agents should be provided.

- *We agree that the claim in comparison to high-end Doppler techniques cannot be made. Demené et al. demonstrated that high-frame-rate Doppler acquisitions with an SVD clutter filter can image vessels down to flow speeds of 2.6 mm/s. We expect ULM and mULM methods to be better tools for the characterization of the microvasculature as they show vessels with even smaller flow speeds below 1 mm/s together with a higher resolution. However, the high-end Doppler techniques have not been applied to tumor characterization so far and thus we reduced the claim by changing the sentence from the abstract "showed superior performance in discriminating tumors with different vascular phenotypes" into*

*"that discriminate tumors with different vascular phenotypes"
(page 2, lines 26-27)*

- *Additionally, we mention the results achieved with high-end Doppler measurements by Demené and coworkers in the discussion section:*

"An interesting alternative reference method for blood velocities might be high-end Doppler methods, which were unfortunately not available to us. Indeed, Demené and coworkers (Demené 2015) impressively demonstrated that these methods are capable to measure flow speeds down to 2.6 mm/s, which is sufficient to

characterize the majority of physiological tissues. However, our experimental angiogenic tumors with their dysfunctional vascularization have a significant fraction of vessels with clearly lower velocity values that would be missed in the analysis and where mULM extends the measurement range to less than 1 mm/s.“
(page 13, lines 270-276)

5. Last sentence is strongly overclaiming with respect to the very low quality of clinical images.

- *We agree and addressed this point by including improved clinical data (see Fig. 6) as well as by reducing our claim in the abstract. The last sentence in abstract has been changed to*

"Furthermore, our initial patient data indicate that mULM can be applied in a clinical ultrasound setting opening avenues for the multiparametric characterization of tumors and the assessment of therapy response."
(page 2, lines 27-29)

6. Introduction: Line 64. Very high frame rates interest is not only to improve detection and tracking but also to acquire much more bubbles locations in a reasonable time and finally to correct for tissue motion artifacts in a robust way. This should be explained as the quality of the final superresolved image is strongly related to: the number of microbubbles correctly detected, the correction of motion artifacts strongly linked to the total acquisition time.

- *We agree with the reviewer's statements and incorporated a better discussion of the advantages of high-frame-rate imaging for ULM. Indeed, more MB locations are measured in the same time and imaging the same MB along their track through the vessel volume allows a more precise mapping of the vasculature. However, for our current application of imaging the smaller vessels in tumors, the quality of images may not suffer too much from the lower frame-rate sampling of positions of the MB: The average speeds of MB are around 0.6 mm/s. For a frame-rate of 50 Hz, the MB move 12 μm on average, which is only 2.4 pixels in our mULM images. Furthermore, the mULM method mitigates the lower number of measurements by exploiting a probabilistic motion model and Kalman-filtering localization information to reduce measurement noise. However, all 2D methods suffer from out-of-plane motion that can only be corrected for in a 3D approach. As high frame-rate volume acquisition with volume rates exceeding 50 Hz is challenging and to date not available in clinical systems, the mULM approach has strong potential to be extended to clinical volume data in the near future.*

The manuscript has been changed as follows:

In the introduction:

"Therefore, Errico and colleagues (Errico 2015) used an experimental imaging system with a very high frame rate (500 frames per second). By detecting the

moving MB more frequently, ambiguous assignments are avoided and the overall detection probability for an MB rises. Additionally, in-plane motion estimation and correction are improved. However, while the detection probability of MBs is increased, the total number of MBs available in the image slice within a given acquisition time are determined by the contrast agent blood concentration and the flow-rate of MBs in the vessels. Thus, the higher frame rates improve the overall image quality and the correct localization of the vessels' course but not the total number of detected vessels. Furthermore, comparable frame rates are not realized in the majority of clinical US systems so far, which makes clinical translation of this method difficult at the moment.”

(page 4, lines 61-70)

In the methods section:

“The comparison of the total number of detections in a given time and image area for the high-frame-rate acquisition of Errico et al. (Errico 2015) (0.17 MB/(frame mm²)) with our experiments (2.6 MB/(frame mm²)) indicates that the detection probability for MB is comparable. Though scanning with the higher frame-rate enables to detect the same MB more often and track them continuously, it may not capture a much higher percentage of MB.”

(page 20, lines 416-421)

7. Results: Resolution was found to be 5 μm . Does it fit with the proposed theoretical analysis by Desailly et al? Could be interesting to compare and refer.

- *This interesting point we are now addressing in our paper: The theoretical limits given by Desailly et al. in (Desailly 2015) would result in localization accuracies for the Vevo MS550D transducer of 79 nm axially and 262 nm laterally (even for SNR of 1.0, correlation of 1.0, center frequency of 40 MHz, and bandwidth of 33 MHz). However, these theoretical limits do not account for several effects that will be present for in vivo data: Inhomogeneities in the tissue's material parameters lead to additional time of flight fluctuations that can be larger than the standard deviation of time of flight measurements as considered by Desailly and coworkers. Additionally, the theoretical limit assumes that isolated MB signals are perfectly isolated with complete removal of the stationary background. Even with good motion compensation techniques, part of the background will still be present leading to speckle disturbing the exact localization of correlation maxima. Finally, the correct superposition of motion corrected MB detections is limited by the accuracy of rigid motion compensation that will not be correct for the complete image region.*
- *The confusion on our statement about the resolution of 5 μm , which was related to the pixel size of the images, has been clarified in the revised manuscript (see paragraph below). Additionally, also in response to reviewer 3, we tried to get a better indication of the actual resolution of our super-resolved vessel imaging. For this, we calculated MB counts on the 5 μm grid and identified locations where smallest vessels were flown through by MB at least 5 times in one of the vessel's pixels (6, 8, 13 times). We rotated the image to cut the vessels perpendicular. The profile was interpolated 8 times to get a smoothed vessel profile for a better*

measurement of the full width half maximum. The FWHM was 10, 9, and 8 μm respectively. These measurements were added in the supplemental material (Supplementary Fig.1) to support the resolution of approximately 10 μm as included in the paper now instead of only the pixel resolution.

- We added the following paragraph in the discussion to clarify the statements on resolution and compare to the much lower limits given by the work of Desailly et al.:

“We chose 5 μm as the pixel size, which is approximately $\lambda/8$ or an eighth of the 40 μm axial extent of the point spread function of the Vevo system in the focus, based on the results of other works using bubble localization: Errico et al. (17) reported an in vivo resolution of $\lambda/6$. Viessmann et al. (40) showed that features 5.1-2.2 times smaller than the point spread function could be resolved by bubble localization with a clinical system in an in vitro setup.

Desailly et al. (Desailly 2015) discussed the lower limits of localization accuracy as estimated from the Cramér-Rao-Lower-Bound. Using their formula results in a lower bound of localization accuracies for the Vevo MS550D transducer of 79 nm axially and 262 nm laterally even for a low SNR of 1.0 and a correlation of 1.0 (center frequency 40 MHz, bandwidth 33 MHz). However, these theoretical limits do not account for several effects that will be present for in vivo data: Inhomogeneities in the tissue’s material parameters lead to additional time of flight fluctuations that can be larger than the standard deviation of time of flight measurements as considered by Desailly and coworkers (Desailly 2015). Additionally, the theoretical limit assumes perfectly isolated bubble signals with complete removal of the stationary background. Even with good motion compensation techniques, part of the background will still be present leading to speckle disturbing the exact localization of correlation maxima. Finally, the correct superposition of motion corrected MB detections is limited by the accuracy of rigid motion compensation that will not be correct for the complete image region. Based on these considerations the resolution was not expected to exceed the 5 μm pixel resolution chosen.

The measurement of the actual resolution without ground truth is difficult. As tracks are mapped with a thickness of the 5 μm pixel resolution, a vessel that is not sampled more than once will seem to be resolved at 5 μm . To get an indication of the resolution of the resulting vessel images, the number of MB passages through a pixel were counted and lateral profiles of small vessels that were passed by at least 5 MB were retrieved. The full-width-half-maximum of the smallest vessels was found to be ca. 10 μm as shown in Supplementary Fig. 1.”

(page 22, lines 468-477; page 23, lines 478-493)

8. Important parameters are missing: the average number of bubbles detected per image, the total number of bubbles detected.

- The information about the average number of bubbles detected per image and the total number of bubble detected per frame have been added:

“In the preclinical measurements, the number of detected MB positions per tumor was $5.1 \cdot 10^4 \pm 3.6 \cdot 10^4$ leading to 32 ± 19 MB per frame with for some tumors highly inhomogeneous spatial distribution”

(page 20, lines 414-416)

"The number of detected MB positions per patient was $1.8 \cdot 10^4 \pm 0.9 \cdot 10^4$, while the number of detected MB per frame was 36 ± 18 ."

(page 28, lines 598-600)

9. Line 368, provide more information about the average and maximal displacement estimated.

- *The information on average and maximal displacement estimates has been added.*

"In the preclinical data sets, the estimated maximum displacement for all frames was $59 \pm 34 \mu\text{m}$, but the maximum displacement of frames which were not excluded was reduced to $20 \pm 16 \mu\text{m}$ with a mean displacement of $9 \pm 10 \mu\text{m}$."

(page 19, lines 399-401)

"In these clinical measurements the maximal displacement of the frames that were not excluded was $147 \pm 26 \mu\text{m}$ with a mean displacement of $114 \pm 34 \mu\text{m}$."

(page 28, lines 597-598)

10. Line 371, Large movements if not compensated do not permit to recover the exact location of the vasculature before motion. This point is key for clinical investigation. The very large difference of image quality between preclinical and clinical results comes from these much more complex configurations. Again, the number of bubbles detected by the algorithm is preclinical and clinical configuration should be given. Mapping the vasculature of tumors or organs requires a very large number of detected bubbles. These parameters should be added and discussed

- *We agree with the reviewer that large movements impede the exact reconstruction of the vasculature. For the preclinical measurements we excluded frames with very large movements, thus ensuring more reliable motion compensation. From the patient examinations we could only use sections of the measurements because of strong out-of-plane movement. For these segments, typical displacement estimates and the number of detections were added (see comments to point 7. and 8. of reviewer 1). The number of detected MB and the influence of measurement times are discussed in the answer to the comment 5 of reviewer 1.*

11. Figure 6 and figure 7 are really not convincing. It is impossible to argue on figure 7 that the decrease of number of detected vessels is due to the chemotherapeutic treatment. The detected bubbles trajectories do not provide a connected vascular network.

- *We included new clinical examinations with more detected MB where we could generate more vessel tracks and therefore reconstruct the tumor vasculature more completely (Fig. 6 a,b). As we already reported, we observed the change in the*

tumor's vascular pattern during the course of the chemotherapy, initially as an increase in the level of the vascularization after the first cycle of the chemotherapy (Supplementary Fig. 4). In our new clinical example (Fig. 6b), we confirmed this finding. Our results suggest that after the first cycle of chemotherapy a vascular decompression effect occurs leading to an increased perfusion of the tumors. Consequently, the delivery of the chemotherapeutic agent to the tumor tissue is enhanced, which leads to pronounced shrinkage of the tumor at subsequent time points. We consider this finding important and plan to substantiate our hypothesis on the vascular decompression effect in future studies.

12. Line 525 provide safety regulations parameters and ethical approval authorization number for the clinical study.

- *The contrast-enhanced ultrasound examination is an approved diagnostic method in examination of breast cancer. The dosage of Sonovue, which is used as a contrast agent in CEUS imaging, as well as the contrast mode settings on the ultrasound device are clinically approved. Mechanical index that was used during examinations was 0.07 and thermal index was below 0.4. CEUS data derived from a clinical study registered at clinicaltrials.gov under the number: NCT03385200.*

Two sentences have been added to the subsection "Study approval" in the "Materials and Methods" section:

"CEUS data derived from a clinical study registered at clinicaltrials.gov under the number: NCT03385200. Mechanical index that was used during examinations was 0.07 and thermal index was below 0.4."

(page 16, line 344; page 17, lines 345-346)

Reviewer 2

1. Very interesting paper developing a new and elegant analysis of vascular heterogeneity of tumors in mice and patients with a tentative of proof of concept.

- *We are excited that the reviewer likes our work and we are thankful for these positive remarks.*

2. The references need to be adapted with more recent and recognized papers as validation of DCE-US.

- *The following references have been added:*
(O'Connor 2017) O'Connor JP, Aboagye EO, Adams JE, Aerts HJ, Barrington SF, Beer AJ, et al. *Imaging biomarker roadmap for cancer studies. Nat Rev Clin Oncol.* 2017;14(3):169-86.
(Lassau 2014) Lassau N, Bonastre J, Kind M, Vilgrain V, Lacroix J, Cuinet M, et al. *Validation of dynamic contrast-enhanced ultrasound in predicting outcomes of antiangiogenic therapy for solid tumors: the French multicenter support for innovative and expensive techniques study. Invest Radiol.* 2014;49(12):794-800.

3. Why use MIOT and not DCE-US which is recognized as standard?

- *We preferred MIOT over DCE-US for the post-processing of the CEUS data because it is well established in our institute for the quantification of relative blood volume (rBV) and proved to provide us with more reliable results in the preclinical set-up than other post-processing techniques (Palmowski 2010). This is reasoned by the fact that because of the small blood volume in mice we can only inject a small volume of contrast agent. Thus, ensuring absolute identical injection speed and duration are difficult, which strongly can influence the peak accumulation of the contrast agent in the tumors. In addition, differences in the blood circulation of the anesthetized animals further influence the contrast agent inflow into the tumors. MIOT postprocessing is less affected by these problems and thus more robust for the assessment of the rBV compared to postprocessing techniques directly on the time-intensity curves (Palmowski 2010). A corresponding paragraph has been added in the Discussion section:*

“Although dynamic-contrast-enhanced ultrasound (DCE-US) with parameter extraction from signal intensity time curves is current standard for the quantification of the patients' data sets (Lassau 2014), we decided to use MIOT for the determination of the rBV since this postprocessing method shows higher robustness and accuracy in the quantification of the level of vascularization in small animals (Palmowski, 2010). This is due to the fact that even small differences in the speed of MB injection and the animals' blood circulation under anesthesia can strongly affect the upslope and the peak of the signal intensity time curve, while it has significantly less impact on the plateau level of the MIOT curve.”

(page 12, lines 243-251)

4. The 3 clinical exams are very heterogeneous in term of machines and settings in order to give some proof of concept

- *We followed the reviewer's suggestion and added new clinical examples. In the current version of the manuscript, all three patients were examined with the same ultrasound device in contrast mode (Fig. 6, Supplemental Fig. 4).*

5. For the references please change

a) number 9 by Imaging biomarker roadmap for cancer studies.

O'Connor JP, Aboagye EO, Adams JE, Aerts HJ, Barrington SF, Beer AJ, Boellaard R, Bohndiek SE, Brady M, Brown G, Buckley DL, Chenevert TL, Clarke LP, Collette S, Cook GJ, deSouza NM, Dickson JC, Dive C, Evelhoch JL, Faivre-Finn C, Gallagher FA, Gilbert FJ, Gillies RJ, Goh V, Griffiths JR, Groves AM, Halligan S, Harris AL, Hawkes DJ, Hoekstra OS, Huang EP, Hutton BF, Jackson EF, Jayson GC, Jones A, Koh DM, Lacombe D, Lambin P, Lassau N, Leach MO, Lee TY, Leen EL, Lewis JS, Liu Y, Lythgoe MF, Manoharan P, Maxwell RJ, Miles KA, Morgan B, Morris S, Ng T, Padhani AR, Parker GJ, Partridge M, Pathak AP, Peet AC, Punwani S, Reynolds AR, Robinson SP, Shankar LK, Sharma RA, Soloviev D, Stroobants S, Sullivan DC, Taylor SA, Tofts PS, Tozer GM, van Herk M, Walker-Samuel S, Wason J, Williams KJ, Workman P, Yankeelov TE, Brindle KM, McShane LM, Jackson A, Waterton JC. Nat Rev Clin Oncol. 2017 Mar;14(3):169-186

- *The reference has been changed as suggested by the reviewer.*

b) number 7 by:

Validation of dynamic contrast-enhanced ultrasound in predicting outcomes of antiangiogenic therapy for solid tumors: the French multicenter support for innovative and expensive techniques study. Lassau N, Bonastre J, Kind M, Vilgrain V, Lacroix J, Cuiet M, Taieb S, Aziza R, Sarran A, Labbe-Devilliers C, Gallix B, Lucidarme O, Ptak Y, Rocher L, Caquot LM, Chagnon S, Marion D, Luciani A, Feutray S, Uzan-Augui J, Coiffier B, Benastou B, Koscielny S

- *The reference has been changed as suggested by the reviewer.*

c) please add this ref just at the end of sentence line 44

Dynamic contrast-enhanced ultrasound parametric maps to evaluate intratumoral vascularization. Pitre-Champagnat S, Leguerney I, Bosq J, Peronneau P, Kiessling F, Calmels L, Coulot J, Lassau N. Invest Radiol. 2015 Apr;50(4):212-7

- *The reference has been added.*

6. for the clinical part: the first clinical case needs to be deleted because no contrast specific mode was used. 3 ml of Sonovue; WHY? This dose is not recommended images are not demonstrative

- *We agree with the Reviewer. The first clinical case has been removed and better examples have been added, which were all recorded with the same clinical ultrasound device in contrast mode (Fig. 6).*

Reviewer 3

1. Tracking of individual microbubbles (ultrasound contrast agent) confined to the vasculature is used to probe the characteristics of the microvascular environment in tumors. The tracking allows to estimate morphological (distance between vessels) as well as functional (velocity and direction of blood flow) information which cannot be obtained using standard ultrasound techniques due to the diffraction limit of the imaging system. Using a preclinical ultrasound system on a mouse model, the authors shows that different tumor types can be discriminated based on information mostly related to the distance between microvessels. Preliminary data obtained during clinical exams are also presented. Overall, this work appears technically sound with novel preclinical and clinical findings, but I would like the authors to perform appropriate revisions and additions.

- *We would like to thank the reviewer for the deep and thorough review and the positive feedback.*

2. Although the term 'super-resolution' is employed throughout the manuscript, I am not sure the technique described here should employ it as simply determining the position of an object to an accuracy greater than the resolution limit for an imaging system is not super-resolution (the imaging bandwidth is not extended beyond the conventional limit).

- *From a technical perspective we completely agree with this concern. The term super-resolution is used correctly when two objects placed closer than the diffraction limit to each other can be resolved when imaged at the same time, e.g. by model-based extrapolation of the bandwidth. Here, we only exploit the high accuracy of localization for single objects that appear in the image at different times, i.e. statistical sampling of the object at higher resolution. For this reason, the term super-localization was proposed as an alternative. However, many publications on these super-localization techniques for microbubble imaging in the last years have nevertheless adopted the term super-resolution for this technique, e.g. these publications cited in the manuscript:*

Errico C, Pierre J, Pezet S, Desailly Y, Lenkei Z, Couture O, et al. Ultrafast ultrasound localization microscopy for deep super-resolution vascular imaging. Nature. 2015;527(7579):499-502.

Christensen-Jeffries K, Browning RJ, Tang MX, Dunsby C, Eckersley RJ. In vivo acoustic super-resolution and super-resolved velocity mapping using microbubbles. IEEE Trans Med Imaging. 2015;34(2):433-40.

Viessmann OM, Eckersley RJ, Christensen-Jeffries K, Tang MX, Dunsby C. Acoustic super-resolution with ultrasound and microbubbles. Physics in medicine and biology. 2013;58(18):6447-58.

Also, the optical techniques based on localization principles, that were developed by the Nobel prize awardees Betzig, Hell, and Moerner, are categorized as "super-resolved fluorescence microscopy".

For these reasons we decided to use the term super-resolution as it seems to be most accepted in the field.

3. In the data presented, individual MBs are localized and tracked to a resolution that is finer than the imaging resolution but the 50 Hz (or lower) frame rate and recording time <1 min excludes a reconstruction depicting the actual vessel size but rather the distance between vessels and blood velocity in the micro vessels. In contrast with refs 16, 17 and 30, the number of detected MBs does not seem to provide a full map of the vasculature giving an estimation of the actual vessel sizes.

- *Unfortunately, it is not easy to compare the number of detections to the cited references. Reference ([16], original reference numbers used) is the only reference where the number of detection events is explicitly mentioned. From these, approximately 13 detection events per frame can be derived. In [30] it is stated, that much less events than in [16] were detected. In [17] the infusion rate was adapted to ensure spatially isolated events. From the infusion rate and the concentration used we would estimate that the number of MB detected per frame was lower than in our experiments.*

However, from the number of detection events, the number of observed MB cannot be derived easily (one observed MB will have multiple detections). On the one hand, the same MB will be detected more often with high than with low frame rates. On the other hand, also the characteristics of the vasculature combined with the imaging plane will influence the ratio of the number of detection events to the number of observed MB: the same MB will be detected more often in a long vessel which is aligned within the imaging plane than in a tortuous vessel entering and leaving the imaging plane. So, we lack the information about the absolute number of MB from the other studies, though it seems that the number of observed MB per time and image area is most likely slightly lower than or similar to our application. Though, the acquisition times differ substantially: In [30], 8000 images per slice were acquired with a frame rate of 500 fps resulting in an acquisition time of 16 s per slice. In [16], 75000 frames were acquired with a frame rate of 500 fps resulting in an acquisition time of 150 s. In [17], the frame rate was 25 fps. The acquisition time could be derived from the infusion information: If the total injected volume did not exceed 200 μ l and the infusion rate was between 0.2-5.0 μ l per minute, the acquisition time must have been around 40-1000 min.

We agree with the reviewer, that we could not reconstruct the full map of the vasculature. In (Dencks 2017) we discussed the compromise to be solved in clinical applications between short measurement times and the reconstruction of the vessel trees. There we showed that our measurement times are sufficient for a reliable estimation of the rBV. This is also supported by our correlative analyses with reference methods (Fig. 6).

We also agree that we do not perfectly determine the vessel sizes for all vessels. In Supplementary Fig. 1 we show that some of the smaller vessels have been flown through by several MB and that in some vessels the size of the vessels could be determined. We agree that this is not possible for all vessels, because part of the vasculature shows only single tracks. However, this is also not clear for the smaller vessels in the images of Errico et al. in [16], as no absolute numbers but

arbitrary units are given in the cuts through the vessels. From our calculations, the number of bubbles per time and area was lower in [16] than in our study:

In [16]: 1,200,000 MB localizations in 74,800 frames measured in 150 ms in an area of approximately 90 mm²: 0.17 MB/(frame mm²), 85 MB/(s mm²).

In our study: A431 tumor, 39 MB per frame in an area of 15 mm²: 2.6 MB/(frame mm²), 130 MB/(s mm²)

To conclude, the MB density may be sufficient to have comparable completeness of the vessel trees as [16], but we agree that from the vessel trees a reliable size estimation is not possible for all vessels in the image. Therefore, we do not use the vessel size as a parameter in our study but rather evaluate the distance to the next track as an estimation of the distance to the next vessel.

We added the Supplementary Fig. 1 and clarified the number of detected bubbles and compared them to the results of Errico et al. in the manuscript. ("Image processing" subsection in "Material and Methods").

"In the preclinical measurements, the number of detected MB positions per tumor was $5.1 \cdot 10^4 \pm 3.6 \cdot 10^4$ leading to 32 ± 19 MB per frame with for some tumors highly inhomogeneous spatial distribution. The comparison of the total number of detections in a given time and image area for the high-frame-rate acquisition of Errico et al. (0.17 MB/(frame mm²)) and our experiments (2.6 MB/(frame mm²)) indicates that the detection probability for MB is comparable. Though scanning with the higher frame-rate enables to detect the same MB more often and track them continuously, it may not capture a much higher percentage of MB."
(page 20, lines 414-421)

(Dencks 2017) Dencks S, Piepenbrock M, Schmitz G, Opacic T, Kiessling F. Determination of adequate measurement times for super-resolution characterization of tumor vascularization. In: Proc. 2017 IEEE International Ultrasonics Symposium (IUS) (2017). doi 10.1109/ULTSYM.2017.8092351

4. The authors claim to achieve an in-plane resolution of 5 μm ("A spatial resolution of approximately 5 μm was achieved", p.5), the manuscript needs to bring more evidence to support the 5 μm resolution obtained, especially with respect to physiological motions over the course of the recording.
 - The resolution was also discussed in the response to question 6. of reviewer 1. The statement about the resolution of 5 μm relates to the pixel size of the images that were generated. This has been clarified in the revised manuscript. Additionally, also in response to reviewer 1, we tried to get a better indication of the actual resolution of our super-resolved vessel imaging. For this, we calculated MB counts on the 5 μm grid and identified locations where smallest vessels were flown through by MB at least 5 times in one pixel (6, 8, 13 times). We rotated the image to cut the vessels perpendicular. The profile was interpolated 8 times to get a smoothed vessel profile for a better measurement of the full width half maximum.

The FWHM was 10, 9, and 8 μm respectively. These measurements were added in the supplemental material (Supplementary Fig. 1) to support the resolution of approximately 10 μm as included in the paper now instead of only the pixel resolution.

We added the following paragraph in the discussion section of the manuscript to clarify the statements on resolution and compare to the much lower limits given by the work of Desailly et al.:

“We chose 5 μm as the pixel size, which is approximately $\lambda/8$ or an eighth of the 40 μm axial extent of the point spread function of the Vevo system in the focus, based on the results of other works using bubble localization: Errico et al. (17) reported an in vivo resolution of $\lambda/6$. Viessmann et al. (40) showed that features 5.1-2.2 times smaller than the point spread function could be resolved by bubble localization with a clinical system in an in vitro setup.

Desailly et al. (Desailly 2015) discussed the lower limits of localization accuracy as estimated from the Cramér-Rao-Lower-Bound. Using their formula results in a lower bound of localization accuracies for the Vevo MS550D transducer of 79 nm axially and 262 nm laterally even for a low SNR of 1.0 and a correlation of 1.0 (center frequency 40 MHz, bandwidth 33 MHz). However, these theoretical limits do not account for several effects that will be present for in vivo data: Inhomogeneities in the tissue’s material parameters lead to additional time of flight fluctuations that can be larger than the standard deviation of time of flight measurements as considered by Desailly and coworkers (Desailly 2015). Additionally, the theoretical limit assumes perfectly isolated bubble signals with complete removal of the stationary background. Even with good motion compensation techniques, part of the background will still be present leading to speckle disturbing the exact localization of correlation maxima. Finally, the correct superposition of motion corrected MB detections is limited by the accuracy of rigid motion compensation that will not be correct for the complete image region. Based on these considerations the resolution was not expected to exceed the 5 μm pixel resolution chosen.

The measurement of the actual resolution without ground truth is difficult. As tracks are mapped with a thickness of the 5 μm pixel resolution, a vessel that is not sampled more than once will seem to be resolved at 5 μm . To get an indication of the resolution of the resulting vessel images, the number of MB passages through a pixel were counted and lateral profiles of small vessels that were passed by at least 5 MB were retrieved. The full-width-half-maximum of the smallest vessels was found to be ca. 10 μm as shown in Supplementary Fig. 1.”

(page 22, lines 468-477; page 23, lines 478-493)

5. What displacement value is considered small or large in the mouse experiments? How much displacement did you measure in the clinical data? Recent articles have emphasized the issue of tissue motion for single MB tracking methods (Hingot et al, Ultrasonics, 2017; Foiret et al., Scientific Reports, 2017), thus it would be interesting to include these papers in the manuscript as it seems motion will be a challenge for the clinical application of the technique.

- *The information about maximal displacement in the preclinical and clinical data as well as the references concerning the necessity and accuracy of motion estimation have been added to the manuscript.*

"In preclinical data sets, the estimated maximum displacement for all frames was $59 \pm 34 \mu\text{m}$, but the maximum displacement of frames which were not excluded was $20 \pm 16 \mu\text{m}$ with a mean displacement of $9 \pm 10 \mu\text{m}$. Similar findings for displacement values were reported by Foiret et al. (Foiret 2017). Hingot and coworkers observed displacements in the rat brain up to $40 \mu\text{m}$ in comparison to the resolution of $8 \mu\text{m}$ under optimal conditions (Hingot 2017). Motion correction on these images was performed at a resolution of $1 \mu\text{m}$ and correction accuracies well below $10 \mu\text{m}$ were considered achievable with the 15 MHz imaging system."

(page 19, lines 399-405)

"In these clinical measurements the maximal displacement of the frames that were not excluded was $147 \pm 26 \mu\text{m}$, with the mean displacement of $114 \pm 34 \mu\text{m}$."

(page 28, lines 597-598)

6. Please provide an estimate of the voxel size for all ultrasound systems used in this work (axial, lateral and elevational resolution) as well as an estimate of the improvement achieved in plane by UBT in the results presented. As 1D arrays are used in this work, it should appear in the text that all distances and velocities are in plane and therefore not absolute as they do not consider the uncertainty in elevation.

- *We included the information of the axial, lateral, and elevational resolution for the Vevo 2100 with the MS550D transducer used in the preclinical studies and the Aplio 500 system with the PLT 1005BT linear array that was used in the clinical study.*

While the information for the Vevo system is publicly available ($40 \mu\text{m}$ axially, $90 \mu\text{m}$ laterally, $200 \mu\text{m}$ elevationally), the information for the Toshiba Aplio were derived from the full width half maximum of single microbubble PSFs from the focal region, the elevational focusing is at 20 mm depth and the elevational resolution of 1 mm was communicated by Canon Medical, former Toshiba Medical ($560 \mu\text{m}$ axially, $780 \mu\text{m}$ laterally, 1 mm elevationally).

"The focal resolution of the preclinical images is $40 \mu\text{m}$ axially, $90 \mu\text{m}$ laterally, and $200 \mu\text{m}$ elevationally."

(page 18, lines 381-383)

"The focal resolution of the clinical images in contrast mode was determined from the full width half maximum of single MB point spread functions at focal depth as $560 \mu\text{m}$ axially and $780 \mu\text{m}$ laterally. The elevational resolution of 1 mm at the elevational focus in 20 mm depth was communicated by Canon Medical Systems."

(page 28, lines 592-596)

- *The resolution of the vessel images in the preclinical case was determined to be approximately $10 \mu\text{m}$ as discussed above. This exceeds the axial and lateral resolution of the preclinical US setup by a factor of 4 and 9, respectively. We*

mention this in the supplementary material (Supplementary Fig. 1), where the resolution is estimated. For the clinical images we cannot give a reliable estimate of the resolution achieved due to the low number of tracks per vessel and the larger slice thickness.

- An additional sentence has been added regarding the in-plane velocities and distance parameters:

"However, it should be noted that all CEUS data are acquired with 1D array transducers and therefore, all distance and velocity parameters are calculated in the imaging plane."

(page 25, lines 523-524)

7. The technique is appealing but I am worried about the physical limitations using 1D arrays. Here, a high-frequency preclinical system dedicated to small animals is used which offers a relatively narrow elevation resolution. However, with clinical systems, as stated the elevation resolution is in the mm range which, combined with motion, may prevent easy analysis. Looking at the clinical data presented, I feel that this is the case (for instance with Fig 6 a, and Fig 7) where MB tracks seem to overlap a lot. The analysis of the UBT parameters on the mouse tumors seem to indicate that the parameters related to the distance between vessels are the most discriminatory, and therefore the translation to human may greatly suffer from the 1D array limitations. If it is the case and 3D imaging is necessary, it mitigates a rapid translation toward clinic as 3D US systems are still sparse.

- *We agree with the reviewer who foresees the problem that comes from the limited assessment in the elevational plane with the 1D arrays and still sparse routine use of matrix transducers, which can offer the solution for this issue. We are also thinking in that direction and we plan in our next step to include more analysis on clinical data in order to get more accuracy in the elevation direction. One possibility could be the reconstruction of the tumor vasculature after collection of the 2D parallel slices during the motorized or hand movement of the 1D arrays over the skin. However, as already discussed in (Lin et al. Theranostics, 2017), certain issues have to be carefully considered in this approach such as acquisition time, motion correction and step size between 2 parallel slices. This is especially challenging when determining the velocity and direction of the individual microbubbles. Another possibility is the use of 2D matrix transducers. As soon as the clear benefit coming from this new generation of transducers is proven, we believe that the production of devices equipped with 2D matrix transducers will increase, which in return may affect the expenses for their production and increase their prevalence in the clinical routine. As a matter of fact, we plan to adapt mULM for the analysis of data acquired with 2D matrix transducers in a clinical setting.*

8. Please consider adding scalebars in Fig 1, 2, 5, 6 and 7 to get a better sense of the tumor dimensions and the MB tracks.

- *We thank the reviewer for this constructive comment. Scale bars have been added in figures 1, 2, 5 and 6. Figure 7 is removed.*

9. In the manuscript, you analyze the distance between vessels but the output the UBT process are individual MB tracks. Does this mean you first group MB tracks that are adjacent to one another and associate them to a vessel? If so, how do you do the association? Do you differentiate the adjacent tracks with the flow direction (for instance adjacent tracks with opposite flow direction)? Do you use the average velocity for each trajectory?

- We agree with the reviewer that the association of the tracks with the actual vessels cannot be solved easily. Thus, in our work, we only measured the distances to the next track. Our rationale is that for tissue blood supply the distance to the next track will have approximately the same value as the distance to the next vessel. Thus, the computation of the distances is based on the tracks, which are reconstructed on the grid of $5 \times 5 \mu\text{m}^2$. Because of this pixel size, it can be assumed that tracks belonging to the same small vessel are placed in the same or neighboring pixels thus having zero distance. Errors might arise from large vessels with distant tracks where small distances are determined between these tracks although the tracks are belonging to the same vessel. However, large vessels are flown through more frequently by MB, most probably filling up these gaps with the additional tracks. In the few cases where this is not true, the error induced by some small - apparent - distances can be neglected. Generally, due to the chaotic vessel network, the definition and computation of distances between two vessels is not possible since they are not parallel but can exhibit any arbitrary angle between each other, also including crossings. Furthermore, the flow direction varies along one vessel, thus impeding a consistent definition based on the flow direction. So, in this step, we did not differentiate adjacent tracks of opposite flow directions. Differentiating vessels of low and of high flow velocities was easily done by preparing two maps in which only tracks of low or of high flow velocities were plotted. Therefore, it was not necessary to come to a decision by averaging the velocities of different tracks.*

A sentence in "Material and Methods" has been added clarifying that we are measuring distances to tracks and not vessels:

"For each pixel, the track distance map provided the shortest distance to the next track, which was used as an estimate for the distance to the closest vessel."

(page 24, lines 505-507)

10. What is the typical number of successive detections for a trajectory?

- The maximum number of successive detections for one trajectory was 42 detections and the typical number is between 2 and 10 detections. This information was added in "Tracking of Microbubbles" in "Materials and Methods":*

"The maximum number of successive detections for one trajectory was 42 detections and the typical number is between 2 and 10 detections."

(page 22, lines 463-464)

11. Regarding Fig 5g, I feel that the comparison between UBT and IHC should be between UBT and μ CT which gives the true 3D distance between vessels in the entire tumor. The imaging plane has an elevation resolution of a few hundred μ m (I am guessing) whereas IHC represents a slice of only 5 μ m, so in a way it is expected to see discrepancies between the two methods. To my eyes, Fig 5a is closer to Fig 5c than Fig 5d and I think a comparison between UBT and μ CT will yield better results.

- *We agree with the reviewer that distances between vessels assessed by μ CT would provide an excellent comparison to the distances measured with UBT. We tried this but, unfortunately, beam hardening artefacts in the μ CT images made proper segmentation of the microvessels difficult and the determination of distances between vessels unreliable. Therefore, we consider the IHC analysis more robust. A sentence has been added to the discussion section to clarify this limitation.*

“Next, the distances obtained by mULM were compared to distances assessed from the histological sections. Even though distances obtained by μ CT would yield more accurate results, since they provide 3D information, the beam-hardening artefacts hindered proper segmentation of the small blood vessels and made the assessment of the distances unreliable. Therefore, distances to the closest vessel calculated from the histological sections were used as reference values. Although the distances calculated from histological specimen had the same trend among the tumor models as those measured by mULM, they were a bit lower. As already discussed for rBV, this can be attributed to the fixation process of the tumor tissue and the loss of interstitial fluids during tumor removal.”

(page 14, lines 277-285)

12. Does the separation between high and low velocities in the analysis of the distances means you predominantly look at the arterial flow for the high velocities and venous flow for the low velocities (my knowledge of tumor vascularization is limited)? Or large versus small vessels? It would be interesting to see separated track maps for the low velocities and for the high velocities.

- *We appreciate this question by the reviewer, which is indeed interesting. In the healthy tissues blood vessels have high hierarchy and are well histologically differentiated in arteries, arterioles, capillaries, venules and veins. This is quite different in tumors where the vascularization is chaotically and heterogeneously distributed, and where vessels often lack the typical wall composition of arteries and veins. Furthermore, there are multiple arterio-venous shunts. Only the big feeding and draining vessels can be classified as arteries and veins with high certainty. Therefore, we decided to just take the median velocity of the tumors as the threshold between high and low flow velocity. We assumed that the arteries are among the fraction of vessels with high flow and with predominant direction towards the tumor core, while the veins dominate the fraction of vessels with low flow and the direction towards the tumor periphery. Furthermore, it can also be expected that low flow vessels are less mature as reported by (Palmowski et al*

2008). To answer this question, we adapted mULM to determine the fraction of the low and high velocities vessels with respect to directions towards the tumor core or periphery.

For each track, the nearest point of the tumor border was taken as the reference point. Then, the maps were computed by binarizing the flow directions into flow away from the reference point (inflow) and towards the reference point (outflow).

We assigned red color to all MB moving towards the tumor center, while all MB that moved to the periphery were displayed in blue color. Then, we created color-coded velocity maps for high and low velocity vessels separately for these two directions. Additionally, the calculated ratios of the area of the low to the high flow velocity vessels in respect to direction towards the tumor core and to the periphery supported our assertions and showed no differences among these parameters. We found that the vascular hierarchies in the tumors are disrupted and that vessels within either the low or the high velocity category cannot be certainly assigned to arteries or veins. Indeed, the blood directions in tumor vasculature are rather chaotic without any dominant direction. We added the Supplementary Fig. 2 as well as a paragraph in the results section to discuss this issue.

“Additionally, we used mULM to determine the inflow and outflow of vessels with high and low flow velocities that would correspond to arteries or veins. We found that this hierarchy, which is normally present in healthy tissues, is disturbed in tumors, where the majority of vessels are angiogenic with many arterio-venous shunts and chaotic directions (Supplementary Fig. 2).”

(page 7, line 129-133)

(Palmowski et al 2008) Palmowski M, Huppert J, Hauff P, Reinhardt M, Schreiner K, Socher MA, Hallscheidt P, Kauffmann GW, Semmler W, Kiessling F. Vessel fractions in tumor xenografts depicted by flow- or contrast-sensitive three-dimensional high-frequency Doppler ultrasound respond differently to antiangiogenic treatment. Cancer Res. 2008 Sep 1;68(17):7042-9.

REVIEWERS' COMMENTS:

Reviewer #1 (Remarks to the Author):

The authors of the manuscript "Motion Model Ultrasound Localization Microscopy for Preclinical and Clinical Multiparametric Tumor Characterization" made an excellent work to improve the initial manuscript.

I revise my initial advice for the removal of the clinical part made in the initial submission review. Although only a limited number of trajectories are obtained here in the clinical setting, the clinical part in the revised manuscript has been strongly improved and could be kept in the manuscript. The clinical images are more convincing, though only a limited number of traces can be detected. Some of the traces obtained in the clinical settings looks like a single segment. Indeed, the number of successive detections for one trajectory is varying between 2 and 10 detections. For the smallest number (2 or 3), it is difficult to be completely sure that the successive detections correspond to the same bubble and to be completely frank, I do not believe that they are in part of the traces. In my opinion, authors should add a sentence to clarify this in the core of the manuscript and highlight this crucial point for future studies. It will help the non-expert reader to understand that this is an important point and should be addressed by our community in the future. It would even be interesting for the reader to have the histogram of the number of detections per trajectory as a supplementary material, and I think the authors should consider this additional figure panel.

The authors made a very good job in answering to my initial list of questions and comments.

Last Minor comments :

Page 6 line 109. "This lack of venous tumor drainage is known to be a typical characteristic of highly angiogenic tumors". This reference demonstrates that the lack of venous tumor drainage induces an important role of mechanical forces during tumor growth. This increase of mechanical forces is known to activate pathways for cancer proliferation (through the beta-catenin and RET pathway) leading to a vicious circle (cancer -> tumor growth-> mechanical forces -> cancer). This was demonstrated recently in Sanchez-Fernandez et al, Nature 2015. It would be interesting to combine this additional reference with ref [19] in order to address more precisely the impact of mechanical forces in the tumor growth.

The complete ref is:

Fernandez-Sanchez ME, Barbier S, Whitehead J, Bealle G, Michel A, Latorre-Ossa H, Rey C, Fouassier L, Claperon A, Brulle L, Girard E, Servant N, Rio-Frio T, Marie H, Lesieur S, Housset C, Gennisson JL, Tanter M, Ménager C, Fre S, Robine S, Farge E: Mechanical induction of the tumorigenic beta-catenin pathway by tumour growth pressure. Nature 2015, 523:92+.

Finally, I would like to congratulate the authors for this excellent work.

Mickael Tanter

Reviewer #3 (Remarks to the Author):

The manuscript has been improved and I support its publication. I look forward to future works on clinical data.

Reviewer #1

(Please note that specifications on the changes in the manuscript reported in this editor letter are related to the pages and lines in the uploaded Word file with tracked changes and that sometimes there is a slight discrepancy with the converted PDF)

1. The authors of the manuscript "Motion Model Ultrasound Localization Microscopy for Preclinical and Clinical Multiparametric Tumor Characterization" made an excellent work to improve the initial manuscript. I revise my initial advice for the removal of the clinical part made in the initial submission review. Although only a limited number of trajectories are obtained here in the clinical setting, the clinical part in the revised manuscript has been strongly improved and could be kept in the manuscript.

- *We thank Professor Tanter for this positive feedback on our work.*

2. The clinical images are more convincing, though only a limited number of traces can be detected. Some of the traces obtained in the clinical settings looks like a single segment. Indeed, the number of successive detections for one trajectory is varying between 2 and 10 detections. For the smallest number (2 or 3), it is difficult to be completely sure that the successive detections correspond to the same bubble and to be completely frank, I do not believe that they are in part of the traces. In my opinion, authors should add a sentence to clarify this in the core of the manuscript and highlight this crucial point for future studies. It will help the non-expert reader to understand that this is an important point and should be addressed by our community in the future. It would even be interesting for the reader to have the histogram of the number of detections per trajectory as a supplementary material, and I think the authors should consider this additional figure panel.

- *We completely agree with this comment. In our algorithm, the tracking of microbubbles is based on a probabilistic model. Therefore, with a certain probability, track continuations can also be wrong. We added the following sentence clarifying this together with the proposed additional Supplementary Figure 5 giving an exemplary histogram of track length distributions for the clinical dataset that is shown in Figure 6a.*

"As mULM builds tracks based on a probabilistic choice, track continuations may also be wrong with a certain probability, which will be higher for short tracks that occur more often in ambiguous situation as in our preliminary clinical data (for a histogram of track lengths see Supplementary Figure 5)."

Added in the Discussion section, page 15, lines 304-307.

3. The authors made a very good job in answering to my initial list of questions and comments.

- *We thank Professor Tanter for the constructive comments that significantly improved our manuscript and we are glad to hear that we accomplished to answer his questions.*

4. Last Minor comments :

Page 6 line 109. "This lack of venous tumor drainage is known to be a typical characteristic of highly angiogenic tumors". This reference demonstrates that the lack of

venous tumor drainage induces an important role of mechanical forces during tumor growth. This increase of mechanical forces is known to activate pathways for cancer proliferation (through the beta-catenin and RET pathway) leading to a vicious circle (cancer -> tumor growth-> mechanical forces -> cancer). This was demonstrated recently in Sanchez-Fernandez et al, Nature 2015. It would be interesting to combine this additional reference with ref [19] in order to address more precisely the impact of mechanical forces in the tumor growth.

The complete ref is:

Fernandez-Sanchez ME, Barbier S, Whitehead J, Bealle G, Michel A, Latorre-Ossa H, Rey C, Fouassier L, Claperon A, Brulle L, Girard E, Servant N, Rio-Frio T, Marie H, Lesieur S, Housset C, Gennisson JL, Tanter M, Ménager C, Fre S, Robine S, Farge E: Mechanical induction of the tumorigenic beta-catenin pathway by tumour growth pressure. Nature 2015, 523:92+.

- *We thank the reviewer for pointing our attention to this very recent reference that we included in the manuscript.*

5. Finally, I would like to congratulate the authors for this excellent work.

- *We are honored to receive this praise from Prof. Tanter, as a highly recognized expert in this field.*

Reviewer #3

1. The manuscript has been improved and I support its publication. I look forward to future works on clinical data.

- *We are highly thankful to the reviewer for the comments that improved our work substantially.*